# XXLTRAFFIC: EXPANDING AND EXTREMELY LONG TRAFFIC FORECASTING BEYOND TEST ADAPTATION

## ABSTRACT

Traffic forecasting is crucial for smart cities and intelligent transportation initiatives, where deep learning has made significant progress in modeling complex spatio-temporal patterns in recent years. However, current public datasets have limitations in reflecting the distribution shift nature of real-world scenarios, characterized by continuously evolving infrastructures, varying temporal distributions, and long temporal gaps due to sensor downtimes or changes in traffic patterns. These limitations inevitably restrict the practical applicability of existing traffic forecasting datasets. To bridge this gap, we present XXLTraffic, **largest available public traffic dataset with the longest timespan collected from Los Angeles, USA, and New South Wales, Australia**, curated to support research in extremely long forecasting beyond test adaptation. Our benchmark includes both typical time-series forecasting settings with hourly and daily aggregated data and novel configurations that introduce gaps and down-sample the training size to better simulate practical constraints. We anticipate the new XXLTraffic will provide a fresh perspective for the time-series and traffic forecasting communities. It would also offer a robust platform for developing and evaluating models designed to tackle the extremely long forecasting problems beyond test adaptation. Our dataset supplements existing spatio-temporal data resources and leads to new research directions in this domain.

## 1 INTRODUCTION

Rapid global population growth and vehicle proliferation have intensified urban traffic congestion. As cities expand and personal transportation reliance grows, strain on road networks leads to longer commutes, higher fuel consumption, and increased emissions. Accurate traffic prediction is vital for intelligent transportation systems, informing strategies to mitigate congestion and enhance mobility through improved route planning and urban development. Effective forecasting requires capturing long-term spatio-temporal relationships in traffic data. Long-term analysis provides context for anomalies in short-term patterns and reveals trends influenced by population cycles, seasonal shifts, and yearly vehicle usage changes. These insights are crucial for developing robust models that adapt to evolving urban traffic dynamics due to demographic and vehicular changes.

In recent years, significant work has focused on both short-term and long-term traffic flow prediction. Deep learning techniques, including Graph Neural Networks (GNNs), have been employed to extract spatial relationships within traffic networks Jin et al. (2023), while Transformer-based architectures have been utilized to capture temporal dependencies over various timescales Shao et al. (2023a). Although these methods have shown promising results, they often rely on datasets that do not fully encapsulate the complexities introduced by rapid population growth and the surging number of vehicles, thus limiting their applicability to real-world scenarios.

There is an emerging need in intelligent transportation systems to design predictive models that extend beyond test adaptation, effectively generalizing to real-world conditions that evolve over time. It is important to note that our concept of 'beyond test adaptation' differs from 'test time adaptation' Guo et al. (2024) as illustrated in Fig 1 that shows the distinctions between them. This shift necessitates models that can handle the multifaceted impacts of demographic changes and vehicle proliferation without relying solely on adaptation to specific test datasets. To achieve this, it is essential to utilize

datasets that accurately represent these evolving conditions over extremely long periods, capturing the intricate patterns influenced by population and vehicular growth.

Motivated by this need, we introduce XXLTraffic, a dataset and framework that expands traffic forecasting beyond test adaptation. By incorporating extremely long-term data, XXL-Traffic better reflects real-world scenarios where traffic patterns are continually affected not just by infrastructure changes like highway construction, but also by shifts in distribution due to factors like population growth and increasing vehicle numbers. We will discuss existing datasets and the specific challenges encountered in establishing XXLTraffic, highlighting how it advances the field by providing a more realistic and comprehensive dataset. This facilitates the development of models capable of adapting to the complexities of real-world traffic dynamics without the limitations of traditional test adaptation approaches.

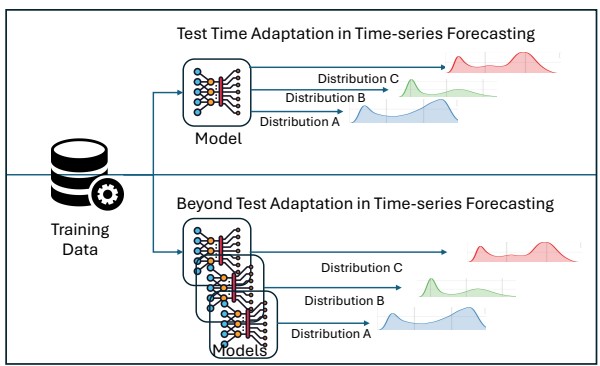

Figure 1: Test-time adaptation in time-series forecasting involves training a single model to fit different test domains, horizons, or gaps. The figure above illustrates this using a gap example. In contrast, the figure below shows our 'beyond test adaptation' where we train separate models for various gap settings.

## 1.1 RECENT ADVANCES IN EXPANDING TRAFFIC DATASETS

Real-world traffic scenarios necessitate more complex prediction settings, involving extended temporal horizons or broader spatial coverage in experiments. In the temporal domain, new settings are typically proposed based on previously published work rather than introducing new datasets: Shao et al. (2023b) and Jia et al. (2024) expanded input and output lengths up to four times on existing datasets. From a spatial perspective, Chen et al. (2021) published a dataset with nodes growing annually and provided an evolving network to support new node predictions. Wang et al. (2023a) proposed a continual learning framework with pattern expansion mechanisms based on Chen et al. (2021). Additionally, SCPT Prabowo et al. (2024) and Large-ST Liu et al. (2024a) offered larger-scale spatial node datasets to support subsequent researchers. Recent work has explored longer temporal step experimental settings and released

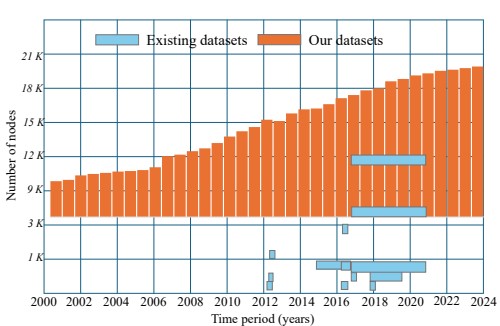

Figure 2: Our dataset is evolving and longer than existing datasets. Existing datasets are either limited by short temporal spans or insufficient spatial nodes. In contrast, our dataset features an evolving growth of spatial nodes and spans over 20 years.

traffic datasets spanning up to five years and thousands of nodes. However, in specific scenarios, such as future traffic prediction for highway planning, these data and experimental settings fall short. As shown in Figure 2, most existing datasets have limitations in temporal span, which inspired us to develop a dataset for expanding and extremely long traffic forecasting. This need for traffic forecasting beyond test adaptation is crucial in various real-world scenarios. For instance, urban planning and infrastructure investment decisions rely heavily on accurate long-term traffic predictions to ensure that developments meet future transportation demands. Commercial real estate site selection and development also depend on knowing future traffic volumes years in advance to optimize location choices and investment strategies. Additionally, governments can formulate more effective environmental policies based on long-term traffic forecasts, such as implementing traffic restrictions or promoting electric vehicles to reduce emissions. These applications highlight the importance of

developing predictive models capable of accurately forecasting traffic patterns over extended periods, facilitating strategic decision-making across multiple sectors. As the temporal span extends, urban infrastructure development and road construction can lead to shifts in traffic patterns, resulting in an evolving domain shift. This observation motivated us to provide an expanding and extremely long traffic dataset. Additionally, the combination of these factors enables the extraction of more temporal patterns from extremely long sequences, allowing for the possibility of longer input sequences.

## 1.2 Challenges

The ultra-dynamic challenge encompasses three key aspects: (1) Continuously evolving states of the underlying spatio-temporal infrastructures, characterized by an expanding number of nodes over the years. This continuous growth introduces complexity as the infrastructure adapts and expands. (2) Evolving temporal distributions over an extremely long observation period, which is crucial for extremely long forecasting beyond different non-contiguous train-test splits. This requires models to adapt to changes in patterns and trends over extensive temporal spans.

We have constructed a traffic dataset with an exceptionally long temporal span and broader regional coverage, providing aggregated data and benchmarking, as well as a benchmarking setup considering extremely long prediction scenarios for future exploration:

- We propose XXLTraffic, a dataset that spans up to 23 years and exhibits evolutionary growth. It includes data from 9 regions, with detailed data collection and processing procedures for expansion and transformation. This dataset supports both temporally scalable and spatially scalable challenges in traffic prediction.

- We present an experimental setup with temporal gaps for extremely long prediction beyong test adaptation and provide a benchmark of aggregated versions of hourly and daily datasets.

- We provide the exploration of input features through evolving temporal distributions over an extremely long observation period. Additionally, our datasets support zero-shot forecasting for new sensors.

## 2 Preliminaries

In this section, we will define traffic data and traffic prediction tasks.

**Definition 1. Traffic datasets:** Traffic data primarily consists of vehicle flow detection data collected by sensors distributed across various locations in the traffic network. It is generally represented by $X_i \in \mathbb{R}^{N \times T \times C}$, where $T$ denotes the time steps, $N$ denotes the number of sensors, and $C$ denotes the number of features.

**Definition 2. Short-term traffic prediction:** Short-term traffic prediction primarily focuses on forecasting traffic speed or flow within the next hour. As shown in Equation 1, the input length $\alpha$ and output length $\beta$ are generally set to 12 steps.

$$[X_{t-(\alpha-1)}, ..., X_{t-1}, X_t] \rightarrow [X_{t+1}, X_{t+2}, ..., X_{t+\beta}], \tag{1}$$

**Definition 3. Long-term multivariate prediction:** This task mainly focuses on long-sequence time series prediction, which includes the traffic dataset. As shown Table 1, the sequence length can reach up to 2880 steps.

**Definition 4. Extremely Long Prediction with Gaps:** Based on Equation 1, the observation and prediction are not adjacent but are instead separated by a gap period $g$, as shown in the following formula.

$$[X_{t-(\alpha-1)}, ..., X_{t-1}, X_t] \rightarrow [X_{t+g+1}, X_{t+g+2}, ..., X_{t+g+\beta}], \tag{2}$$

## 3 Gaps and Comparison with Existing Traffic Datasets

As shown in Table 1, existing traffic prediction work can easily be divided into short-term and long-term settings. The short-term setting originated from the STGCNYu et al. (2018) work, while

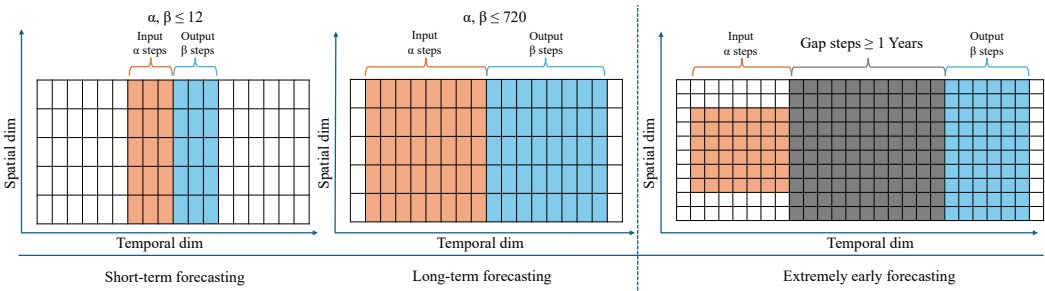

Figure 3: Our expanding and extremely long prediction addresses the existing limitations in both short-term and long-term predictions.

the long-term setting was first introduced by LSTNet Lai et al. (2018) and subsequently established as a widely adopted experimental framework by Informer Zhou et al. (2021). In recent years, short-term prediction typically has a maximum step length of 12 steps, while long-term prediction reaches up to 720 steps. However, works such as Witran Jia et al. (2024) and DAN Li et al. (2024) recognized the need for even longer step predictions in practical applications, extending the length to a maximum of four times the typical length. Despite the differences in step lengths, their observed and predicted values are concatenated tightly together, as shown in Equation 1. To accommodate complex real-world scenarios, such as highway route planning predictions, it is necessary to introduce a gap of several years between observation and prediction. Typically, existing datasets lack the temporal coverage required to support gaps exceeding one year.At the same time, predicting several years in advance also implies the need to forecast traffic for sensors at new locations, taking into account the evolving nature of the road network. Even when such coverage is available, works like Wang et al. (2023a) and Chen et al. (2021) utilize evolving datasets but do not provide sufficient data to train models for extended durations. To overcome these gaps, our Expanding and Extremely Long Traffic Dataset robustly supports these complex scenarios.

Table 1: Summary of recent short-term traffic forecasting and long-term multivariate forecasting

| Datasets | Model | Series Length |
|---|---|---|
| Short-term | STGCN (Yu et al., 2018) | {3,6,9,12} |
| | DCRNN (Li et al., 2018), GWN (Wu et al., 2019), BTF (Chen & Sun, 2021), DMSTGCN (Han et al., 2021), GTS (Shang et al., 2021), STGODE (Fang et al., 2021), PM-MemNet (Lee et al., 2021), STAEFormer (Liu et al., 2023) | {3,6,12} |
| | AGCRN (Bai et al., 2020), STSGCN (Song et al., 2020), ,DSTAGNN Lan et al. (2022), D2STGNN (Shao et al., 2022), DyHSL (Zhao et al., 2023),PDFormer (Jiang et al., 2023), MultiSPANS Zou et al. (2024), GMSDR (Liu et al., 2022) | {12} |
| Long-term | MTGNN (Wu et al., 2020b),LSTNet (Lai et al., 2018) | {3,6,12,24} |
| | ARU (Deshpande & Sarawagi, 2019) | {12,24,48,168,336} |
| | LogSparse_Trans (Li et al., 2019) | {24,48,72,96,120,144,168,192} |
| | AST (Wu et al., 2020a) | {8,24,168,336} |
| | SSDNet (Lin et al., 2021) | {20,24,30,138} |
| | Informer (Zhou et al., 2021), Autoformer (Wu et al., 2021), FEDformer (Zhou et al., 2022), Linear (Li et al., 2022), Triformer (Cirstea et al., 2022), Pyraformer (Liu et al., 2021) DSformer (Yu et al., 2023),DeepTime (Woo et al., 2023),DLinear (Zeng et al., 2023) | {24,48,96,192,336,720} |
| | Witran (Jia et al., 2024) | {168, 336, 720, 1440, 2880} |
| | DAN (Li et al., 2024) | {288, 672, 1440} |

## 4 THE XXLTRAFFIC DATASETS

### 4.1 DATA COLLECTION

We obtained the expanding and extremely long traffic sensor data from the California Department of Transportation (CalTrans) Performance Measurement System[1] (PeMS) Chen et al. (2001) and Transport for NSW[2]. PeMS is an online platform that collects traffic data from 19,561 sensors

---

[1]https://pems.dot.ca.gov/

[2]https://maps.transport.nsw.gov.au/egeomaps/traffic-volumes/index.html#/?z=6

distributed across California state highways. These sensor locations are divided into nine districts. We downloaded all the raw data for these nine districts from the initial data release up to March 20, 2024. The system automatically generates a daily data file for each district, containing data from all sensors within each district. We have stored the complete raw data files in an open-source repository for quick access, which will be released after the publication. The tfNSW is an open-source data platform provided by Transport for NSW, featuring traffic flow data collected from sensors distributed along major roads throughout the state of New South Wales of Australia. The data is available at a minimum granularity of one hour.

Table 2: Comparison between our XXLTraffic dataset and the existing traffic datasets.

| Reference | Dataset | Samples | Nodes | Time Interval | Time Span | Time Period |
|---|---|---|---|---|---|---|
| DCRNN | **METR-LA** | 34,272 | 207 | 5 mins | 4 months | 03/2012 - 06/2012 |
| | **PEMS-BAY** | 52,116 | 325 | 5 mins | 6 months | 01/2017 - 05/2017 |
| LSTNet | **Traffic** | 17,544 | 862 | 1 hour | 2 years | 01/2015 - 12/2016 |
| STGCN | **PEMSD7(M)** | 12,672 | 228 | 5 mins | 2 months | 05/2012 - 06/2012 |
| | **PEMSD7(L)** | 12,672 | 1026 | 5 mins | 2 months | 05/2012 - 06/2012 |
| ASTGCN | **PEMSD4-I** | 17,002 | 228 | 5 mins | 2 months | 01/2018 - 02/2018 |
| | **PEMSD8-I** | 17,856 | 1,979 | 5 mins | 2 months | 07/2016 - 08/2016 |
| STSGCN | **PEMS03** | 26,208 | 358 | 5 mins | 11 months | 01/2018 - 11/2018 |
| | **PEMS04** | 16,992 | 307 | 5 mins | 2 months | 01/2018 - 02/2018 |
| | **PEMS07** | 28,224 | 883 | 5 mins | 2 months | 05/2017 - 06/2017 |
| | **PEMS08** | 17,856 | 170 | 5 mins | 2 months | 07/2016 - 08/2016 |
| Large-ST | **CA** | 525,888 | 8,600 | 5 mins | 5 years | 01/2017 - 12/2021 |
| | **GLA** | 525,888 | 3,834 | 5 mins | 5 years | 01/2017 - 12/2021 |
| | **GBA** | 525,888 | 2,352 | 5 mins | 5 years | 01/2017 - 12/2021 |
| | **SD** | 525,888 | 716 | 5 mins | 5 years | 01/2017 - 12/2021 |
| Ours | **Full_PEMS03** | 2,419,488 | 1809 | 5 mins | 23.00 years | 03/2001 - 03/2024 |
| | **Full_PEMS04** | 2,287,872 | 4,089 | 5 mins | 21.75 years | 06/2002 - 03/2024 |
| | **Full_PEMS05** | 1,998,720 | 573 | 5 mins | 19.00 years | 03/2005 - 03/2024 |
| | **Full_PEMS06** | 1,945,728 | 705 | 5 mins | 18.50 years | 09/2005 - 03/2024 |
| | **Full_PEMS07** | 2,287,872 | 4,888 | 5 mins | 21.75 years | 06/2002 - 03/2024 |
| | **Full_PEMS08** | 2,419,488 | 2,059 | 5 mins | 23.00 years | 03/2001 - 03/2024 |
| | **Full_PEMS10** | 1,998,720 | 1,378 | 5 mins | 19.00 years | 03/2005 - 03/2024 |
| | **Full_PEMS11** | 2,261,376 | 1,440 | 5 mins | 21.50 years | 09/2002 - 03/2024 |
| | **Full_PEMS12** | 2,331,360 | 2,587 | 5 mins | 22.16 years | 01/2002 - 03/2024 |
| | **tfNSW** | 100,056 | 27 | 60 mins | 11.42 years | 01/2013 - 05/2024 |

As illustrated in Table 2, our collected dataset significantly exceeds existing datasets in terms of both temporal coverage and the number of spatial nodes. The dataset sample will be available on: https://anonymous.4open.science/r/XXLTraffic-F281, which includes the raw data, sensor meta-data (containing sensor IDs, geographical coordinates, associated road information, etc.), the data processing pipeline code, and the processed datasets.

## 4.2 DATA PREPROCESSING

Based on the 23 years of raw data we collected, we conducted rigorous data filtering and aggregation. The PeMS system has continuously evolved, expanding from a few sensors in 2001 to over 4,000 sensors in some districts today. To support our setting of extremely long forecasting with gaps, we selected a subset of sensors that were installed in the early stages and have consistently collected new data up to the present(named `gap dataset`), which is shown in the Appendix. This extensive `gap dataset` effectively underpins the extremely long forecasting with gaps demonstrated in Figure 3. Utilizing the `gap dataset`, we performed both `hourly` and `daily` aggregations, which will be employed for gap-free long-term forecasting benchmarking. We will provide standard long-term forecasting benchmarks for both the `hourly` and `daily` datasets.

## 4.3 DATA OVERVIEW

The XXLTraffic dataset is distributed across highways in the state of California, as illustrated in the Figure 4a. The nine colors represent nine districts. From Figures 4b, 4c, and 4d, we can

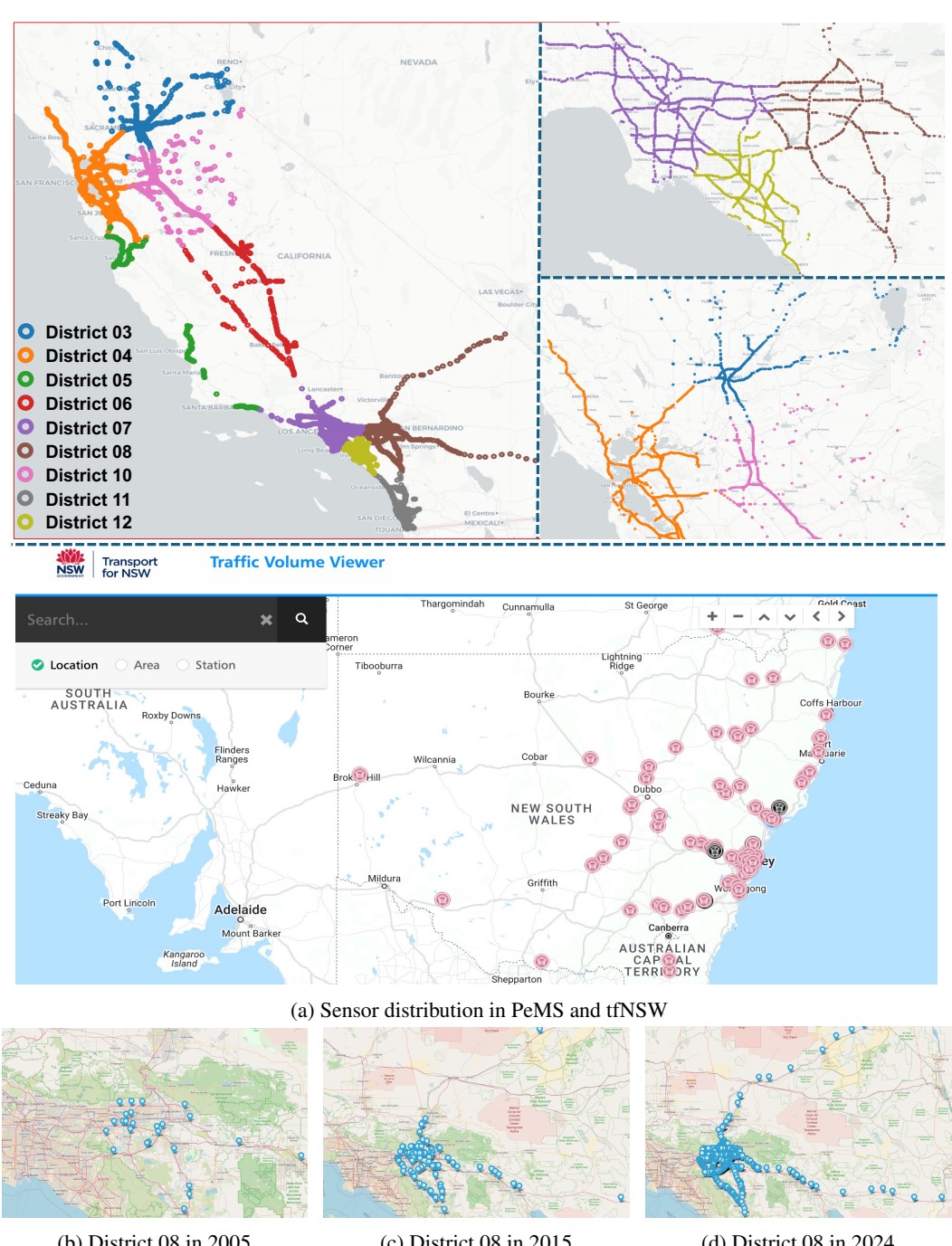

(a) Sensor distribution in PeMS and tfNSW

(b) District 08 in 2005          (c) District 08 in 2015          (d) District 08 in 2024

Figure 4: XXLTraffic dataset overview and its evolving development. This figure provides a global overview and two local overviews, showcasing the diversity of sensor distribution. The lower parts highlights a selected region to illustrate the growth and changes in traffic sensors over time.

clearly observe the evolutionary growth of the sensors. The sensors are extensively distributed across both urban and suburban areas, offering diverse modalities. Additionally, the sensors are densely interconnected, enabling the formation of a high-quality traffic graph dataset.

It is evident that sensors at the same location may collect completely different distributions over the course of urban evolution. As shown in Figure 5, some sensors have maintained the same distribution

from 2005 to 2024, while others have experienced significant changes in distribution. The temporal changes causing domain shifts present a significant challenge for our extremely long forecasting.

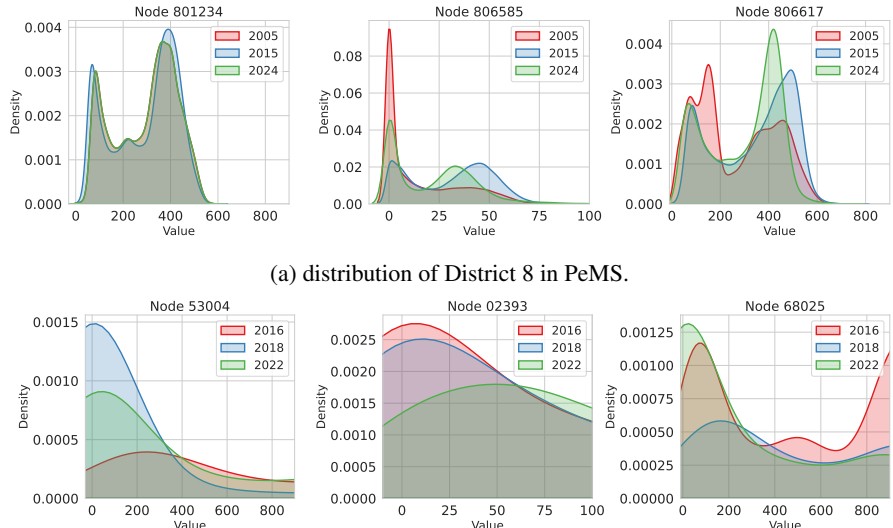

(a) distribution of District 8 in PeMS.

(b) distribution of tfNSW.

Figure 5: Sensor traffic status distribution of District 8 in PeMS from 2005 to 2024 in 5a and from 2016 to 2022 in NSW in 5b. While some sensors exhibit minimal changes, others show significant distribution differences, regardless of whether they are in low-traffic or high-traffic areas. This presents substantial challenges for extremely long forecasting with long gaps.

### 4.4 XXLTRAFFIC LICENCE

The XXLTraffic dataset is licensed under CC BY-NC 4.0 International: https://creativecommons.org/licenses/by-nc/4.0. Our code is available under the MIT License: https://opensource.org/licenses/MIT. Please check the official repositories for the licenses of any specific baseline methods used in our codebase.

## 5 EXPERIMENTS

We conducted experiments for both extremely long forecasting with gaps using `gap dataset` and conventional long-term forecasting using `hourly dataset` and `daily dataset`. Additionally, referring to the definition in Figure 3, we set the gap parameter $g$ as 1 year, 1.5 years, and 2 years for the `gap dataset`, as illustrated by Figure 6.

### 5.1 DATASETS

We conducted experiments on all proposed sub-datasets. To maintain consistency with previous state-of-the-art benchmarks, we selected districts 03, 04, and 08 (widely recognized as PEMS03/04/08) for the experiments using the gap, and districts 03, 04,07 and 08 for hourly, and daily experiments. Results for other datasets are presented in Appendix. All sub-datasets were divided into training, validation, and test sets using a 6:2:2 ratio. For the `gap dataset`, due to the extensive span of up to 20 years resulting in a large sample size, we fixed a seed during data preprocessing to select 10% of the dataset for training and testing to quickly demonstrate our results. The details of the datasets used in our benchmarking is in Appendix A.1.

### 5.2 BASELINES

In our comparison experiments, we adopted four popular baselines, including MLP, Transformer, and Mamba architectures. Informer Zhou et al. (2021) introduces an efficient transformer for long

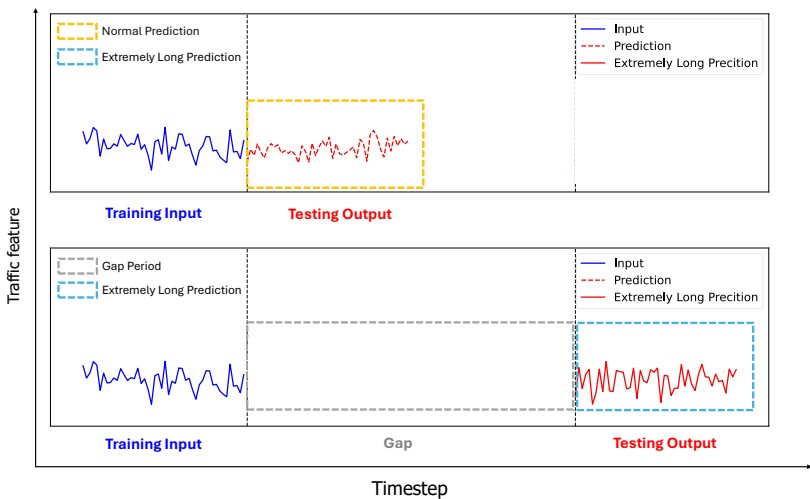

Figure 6: Problem definition. The yellow boxes represent typical predictions, the gray boxes denote gap periods between observation and prediction, and the blue boxes indicate extended predictions.

sequence time-series forecasting using ProbSparse self-attention and self-attention distilling, enabling encoder-decoder architectures to handle long sequences effectively. MICN Wang et al. (2023b) proposes a multi-scale context network that models both local and global contexts for long-term time series forecasting, capturing patterns across different temporal scales to enhance performance. FEDformer Zhou et al. (2022) introduces a frequency-enhanced decomposed transformer that models time series in both time and frequency domains, improving long-term forecasting by effectively capturing temporal patterns. PatchTST Nie et al. applies transformers to time series by treating them as sequences of patches, enabling effective long-term forecasting through self-attention over patch representations to capture temporal dependencies. Autoformer Wu et al. (2021), an earlier state-of-the-art model, leverages a decomposition architecture and auto-correlation mechanism to enhance efficiency and accuracy in long-term time series forecasting, outperforming traditional Transformer models. iTransformer Liu et al. (2024b) is the latest and most effective Transformer-based model, utilizing attention and feed-forward networks on inverted dimensions, embedding time points into variate tokens. DLinear Zeng et al. (2023) challenges the effectiveness of Transformer models by proposing a simple one-layer linear model that captures temporal relations in an ordered set of continuous points. It employs positional encoding and uses tokens to embed sub-series, preserving some ordering information in Transformers. Lastly, Mamba Gu & Dao (2023), a well-known sequential model from last year, uses a bidirectional Mamba block to extract inter-variate correlations and temporal dependencies. Additionally, we have selected five SOTA baselines Yu et al. (2018); Guo et al. (2019); Wu et al. (2019); Bai et al. (2020); Jiang et al. (2023) from traffic forecasting domain.

### 5.3 IMPLEMENTATION DETAILS

We adopted the default settings of the Time-Series-Library Wu et al. (2022) to conduct a comprehensive comparison of baselines. We use the results of five random seeds as the average. We used 96 time steps as input and 336 time steps as ground truth. The code was implemented in PyTorch and executed on a V100 GPU with 32GB memory and 384GB RAM, provided by NCI Australia, an NCRIS-enabled capability supported by the Australian Government.

### 5.4 RESULTS OF EXTREMELY LONG FORECASTING WITH GAPS

We use Mean Squared Error (MSE) and Mean Absolute Error (MAE) metrics to evaluate performance, averaging results across different seeds. It is observed that nearly all results are poor, highlighting the significant challenge posed by domain shifts over time for extremely long forecasting with gaps. These baseline results also indicate that traditional the-state-of-the-art (SOTA) rankings and methodologies

Table 3: Comparison in gap dataset. The bold text indicates the best.

| Gap Data | Gap | Metric | Horizon | Mamba | iTrans | DLinear | Autof | Infor | FEDFo | MICN | Patch | STGCN | ASTGCN | GWN | AGCRN | PDFor |
|---|---|---|---|---|---|---|---|---|---|---|---|---|---|---|---|---|
| PEMS03 | 1-year | MSE | 96 | 1.457 | 1.636 | 1.500 | 1.301 | 0.673 | 0.934 | **0.514** | 0.803 | 0.556 | 0.765 | 0.676 | 0.596 | 0.621 |
| | | | 192 | 1.472 | 1.597 | 1.542 | 1.266 | 0.739 | 0.960 | **0.528** | 0.852 | 0.562 | 0.764 | 0.581 | 0.565 | 0.545 |
| | | | 336 | 1.434 | 1.512 | 1.531 | 1.137 | 0.699 | 0.849 | **0.493** | 0.825 | 0.561 | 0.717 | 0.582 | 0.580 | 0.574 |
| | | MAE | 96 | 0.913 | 0.989 | 0.933 | 0.906 | 0.552 | 0.697 | **0.515** | 0.647 | 0.536 | 0.618 | 0.574 | 0.562 | 0.576 |
| | | | 192 | 0.922 | 0.970 | 0.945 | 0.867 | 0.581 | 0.715 | **0.513** | 0.664 | 0.539 | 0.626 | 0.546 | 0.543 | 0.536 |
| | | | 336 | 0.913 | 0.935 | 0.935 | 0.807 | 0.560 | 0.640 | **0.491** | 0.636 | 0.538 | 0.598 | 0.543 | 0.552 | 0.548 |
| | 1.5-year | MSE | 96 | 1.485 | 1.879 | 1.653 | 1.467 | 1.138 | 1.190 | **0.839** | 1.245 | 1.256 | 1.441 | 1.250 | 1.168 | 1.256 |
| | | | 192 | 1.442 | 1.753 | 1.642 | 1.266 | 1.276 | 1.374 | **0.844** | 1.339 | 1.187 | 1.395 | 1.194 | 1.122 | 1.117 |
| | | | 336 | 1.446 | 1.662 | 1.632 | 1.137 | 1.340 | 1.259 | **0.755** | 1.359 | 1.167 | 1.273 | 1.015 | 1.176 | 1.182 |
| | | MAE | 96 | 0.942 | 1.078 | 0.976 | 0.945 | 0.763 | 0.795 | **0.686** | 0.863 | 0.884 | 0.959 | 0.875 | 0.866 | 0.899 |
| | | | 192 | 0.934 | 1.017 | 0.968 | 0.867 | 0.827 | 0.715 | **0.681** | 0.906 | 0.872 | 0.947 | 0.856 | 0.843 | 0.842 |
| | | | 336 | 0.938 | 0.987 | 0.968 | 0.807 | 0.845 | 0.640 | **0.632** | 0.903 | 0.853 | 0.877 | 0.832 | 0.862 | 0.871 |
| | 2-year | MSE | 96 | 1.359 | 1.844 | 1.568 | 1.328 | 1.642 | **1.205** | 1.359 | 1.817 | 2.059 | 2.236 | 1.987 | 1.950 | 2.068 |
| | | | 192 | **1.216** | 1.729 | 1.473 | 1.235 | 1.955 | 1.489 | 1.308 | 1.899 | 2.055 | 2.180 | 1.877 | 1.746 | 1.897 |
| | | | 336 | 1.294 | 1.614 | 1.407 | 1.220 | 1.982 | 1.615 | **0.966** | 1.656 | 2.022 | 1.889 | 1.847 | 1.930 | 1.982 |
| | | MAE | 96 | 0.833 | 1.048 | 0.954 | 0.894 | 0.956 | **0.816** | 0.911 | 1.104 | 1.176 | 1.234 | 1.148 | 1.150 | 1.189 |
| | | | 192 | **0.772** | 1.008 | 0.896 | 0.834 | 1.067 | 0.933 | 0.866 | 1.119 | 1.184 | 1.213 | 1.117 | 1.070 | 1.131 |
| | | | 336 | 0.801 | 0.960 | 0.870 | 0.838 | 1.086 | 0.978 | **0.700** | 1.016 | 1.169 | 1.117 | 1.107 | 1.140 | 1.161 |
| PEMS04 | 1-year | MSE | 96 | 1.325 | 1.644 | 1.396 | 0.819 | **0.624** | 0.712 | 0.721 | 1.309 | 1.233 | 1.311 | 1.310 | 1.290 | 1.101 |
| | | | 192 | 1.438 | 1.587 | 1.440 | 0.941 | 0.694 | 0.679 | **0.611** | 1.358 | 1.398 | 1.443 | 1.255 | 1.318 | 1.078 |
| | | | 336 | 1.424 | 1.447 | 1.421 | 0.853 | 0.668 | 0.584 | **0.526** | 1.338 | 1.505 | 1.325 | 1.292 | 1.302 | 1.085 |
| | | MAE | 96 | 0.914 | 1.037 | 0.955 | 0.717 | **0.593** | 0.650 | 0.632 | 0.917 | 0.904 | 0.929 | 0.870 | 0.865 | 0.851 |
| | | | 192 | 0.642 | 1.018 | 0.965 | 0.767 | 0.634 | 0.638 | **0.586** | 0.932 | 1.025 | 1.023 | 0.879 | 0.907 | 0.816 |
| | | | 336 | 0.935 | 0.960 | 0.954 | 0.730 | 0.615 | 0.592 | **0.534** | 0.916 | 1.104 | 0.938 | 0.865 | 0.899 | 0.821 |
| | 1.5-year | MSE | 96 | 1.204 | 2.014 | 1.488 | 0.981 | **0.618** | 0.664 | 0.638 | 1.171 | 1.369 | 1.350 | 1.684 | 1.222 | 1.097 |
| | | | 192 | 0.982 | 1.649 | 1.301 | 0.867 | 0.622 | 0.679 | **0.570** | 1.109 | 1.642 | 1.549 | 1.501 | 1.346 | 1.214 |
| | | | 336 | 0.961 | 1.352 | 1.298 | 0.762 | 0.646 | 0.488 | **0.482** | 1.158 | 1.368 | 1.199 | 1.584 | 0.231 | 1.130 |
| | | MAE | 96 | 0.890 | 1.193 | 0.995 | 0.779 | 0.592 | 0.632 | **0.581** | 0.870 | 1.082 | 0.937 | 1.063 | 0.961 | 0.849 |
| | | | 192 | 0.787 | 1.038 | 0.913 | 0.735 | 0.588 | 0.638 | **0.547** | 0.846 | 1.035 | 1.075 | 0.988 | 0.989 | 0.903 |
| | | | 336 | 0.792 | 0.929 | 0.908 | 0.679 | 0.598 | 0.527 | **0.494** | 0.850 | 0.862 | 0.832 | 0.997 | 0.963 | 0.854 |
| | 2-year | MSE | 96 | 1.220 | 1.652 | 1.446 | 0.909 | **0.650** | 0.685 | 0.666 | 1.284 | 1.653 | 1.247 | 1.669 | 1.236 | 1.099 |
| | | | 192 | 1.004 | 1.189 | 1.268 | 0.909 | 0.639 | 0.621 | **0.596** | 1.151 | 1.545 | 1.356 | 1.554 | 1.269 | 1.159 |
| | | | 336 | 1.198 | 1.584 | 1.269 | 0.898 | 0.717 | 0.521 | **0.481** | 1.205 | 1.556 | 1.174 | 1.128 | 1.314 | 1.032 |
| | | MAE | 96 | 0.893 | 1.074 | 0.977 | 0.755 | **0.604** | 0.644 | 0.609 | 0.913 | 1.074 | 0.894 | 1.057 | 0.901 | 0.861 |
| | | | 192 | 0.807 | 0.870 | 0.893 | 0.747 | 0.601 | 0.607 | **0.570** | 0.867 | 1.023 | 0.948 | 1.041 | 0.915 | 0.891 |
| | | | 336 | 0.864 | 1.016 | 0.898 | 0.736 | 0.640 | 0.552 | **0.502** | 0.881 | 1.011 | 0.842 | 0.869 | 0.947 | 0.816 |
| PEMS08 | 1-year | MSE | 96 | 5.411 | 1.514 | 1.771 | **1.153** | - | 1.636 | 1.556 | 1.926 | 2.531 | 3.119 | 2.185 | 2.158 | 1.921 |
| | | | 192 | 12.620 | 1.499 | 1.762 | **1.202** | - | 1.401 | 1.408 | 1.887 | 2.560 | 2.405 | 2.223 | 2.144 | 2.248 |
| | | | 336 | 9.614 | 1.654 | 1.961 | 1.184 | - | 1.870 | **1.093** | 1.962 | 2.482 | 2.086 | 2.228 | 2.036 | 1.994 |
| | | MAE | 96 | 1.319 | 0.950 | 1.058 | **0.843** | - | 1.384 | 0.979 | 1.104 | 1.325 | 1.404 | 1.280 | 1.220 | 1.148 |
| | | | 192 | 1.370 | 0.900 | 1.059 | **0.845** | - | 0.877 | 0.918 | 1.106 | 1.337 | 1.276 | 1.311 | 1.213 | 1.260 |
| | | | 336 | 1.378 | 0.984 | 1.131 | 0.849 | - | 1.078 | **0.761** | 1.133 | 1.317 | 1.181 | 1.312 | 1.185 | 1.179 |
| | 1.5-year | MSE | 96 | 4.453 | 2.286 | 1.978 | 1.428 | - | 1.389 | **1.034** | 1.362 | 1.772 | 2.370 | 1.309 | 1.166 | 1.664 |
| | | | 192 | 9.413 | 1.713 | 1.606 | 1.314 | - | 1.179 | **1.039** | 1.666 | 1.522 | 1.730 | 1.074 | 1.144 | 1.518 |
| | | | 336 | 10.457 | 1.890 | 1.736 | 1.320 | - | 1.197 | **0.868** | 1.272 | 1.495 | 1.444 | 1.533 | 1.078 | 1.049 |
| | | MAE | 96 | 1.061 | 1.196 | 1.072 | 0.901 | - | 0.871 | **0.758** | 0.867 | 1.084 | 1.192 | 0.896 | 0.808 | 1.055 |
| | | | 192 | 1.046 | 0.971 | 0.928 | 0.833 | - | 0.768 | **0.755** | 1.001 | 0.979 | 1.043 | 0.805 | 0.810 | 1.012 |
| | | | 336 | 1.063 | 1.488 | 0.985 | 0.836 | - | 0.782 | **0.680** | 0.853 | 0.966 | 0.948 | 0.999 | 0.784 | 0.801 |
| | 2-year | MSE | 96 | 5.117 | 2.400 | 1.969 | 1.474 | - | 1.494 | **1.030** | 1.393 | 1.336 | 1.962 | 1.789 | 1.393 | 1.071 |
| | | | 192 | 12.769 | 1.974 | 1.670 | 1.505 | - | 1.292 | **1.044** | 1.218 | 1.292 | 1.292 | 1.007 | 1.229 | 1.171 |
| | | | 336 | 13.382 | 1.936 | 1.628 | 1.464 | - | 1.253 | **0.904** | 1.246 | 1.181 | 1.227 | 1.181 | 1.246 | 1.003 |
| | | MAE | 96 | 0.953 | 1.232 | 1.069 | 0.895 | - | 0.885 | **0.740** | 0.856 | 0.868 | 1.069 | 1.017 | 0.856 | 0.792 |
| | | | 192 | 0.933 | 1.070 | 0.942 | 0.883 | - | 0.795 | **0.741** | 0.813 | 0.819 | 0.795 | 0.761 | 0.813 | 0.826 |
| | | | 336 | 0.946 | 1.056 | 0.926 | 0.874 | - | 0.772 | **0.685** | 0.814 | 0.806 | 0.842 | 0.813 | 0.814 | 0.768 |
| tfNSW | 1-year | MSE | 96 | 1.480 | 1.236 | 1.166 | 1.320 | 1.354 | 1.377 | 1.219 | **0.946** | 1.176 | 1.404 | 1.451 | 1.283 | 1.070 |
| | | | 192 | 1.543 | 1.088 | 1.184 | 1.599 | 1.229 | 1.312 | 1.262 | **1.026** | 1.495 | 1.547 | 1.580 | 1.267 | 1.019 |
| | | | 336 | 1.459 | 1.099 | 1.196 | 1.477 | 1.242 | 1.219 | 1.366 | 0.949 | 1.617 | 1.534 | | 1.048 | **0.822** |
| | | MAE | 96 | 0.845 | 0.822 | **0.748** | 0.879 | 0.777 | 0.895 | 0.778 | 0.778 | 0.776 | 0.874 | 0.871 | 0.824 | 0.756 |
| | | | 192 | 0.867 | **0.733** | 0.758 | 0.961 | 0.757 | 0.890 | 0.799 | 0.821 | 0.880 | 0.889 | 0.904 | 0.828 | 0.753 |
| | | | 336 | 0.847 | 0.739 | 0.763 | 0.934 | 0.776 | 0.838 | 0.833 | 0.774 | 0.901 | 0.910 | 0.892 | 0.786 | **0.710** |
| | 1.5-year | MSE | 96 | 1.745 | 1.355 | 1.372 | 1.658 | 1.615 | 1.440 | 1.204 | **1.025** | 1.220 | 1.286 | 1.598 | 1.249 | 1.153 |
| | | | 192 | 1.784 | 1.260 | 1.367 | 1.666 | 1.484 | 1.452 | 1.251 | **1.058** | 1.624 | 1.615 | 1.620 | 1.228 | 1.095 |
| | | | 336 | 1.628 | 1.284 | 1.385 | 1.515 | 1.500 | 1.346 | 1.203 | **0.995** | 1.621 | 1.621 | 1.220 | 1.163 | 1.049 |
| | | MAE | 96 | 0.944 | 0.877 | 0.846 | 1.016 | 0.879 | 0.919 | **0.778** | | 0.785 | 0.813 | 0.940 | 0.809 | 0.790 |
| | | | 192 | 0.968 | 0.818 | 0.846 | 0.997 | 0.854 | 0.932 | **0.789** | 0.823 | 0.934 | 0.902 | 0.950 | 0.808 | 0.792 |
| | | | 336 | 0.939 | 0.854 | 0.854 | 0.945 | 0.883 | 0.871 | **0.770** | 0.788 | 0.900 | 0.900 | | 0.814 | 0.811 |
| | 2-year | MSE | 96 | 1.364 | 1.019 | 1.008 | 1.319 | 1.101 | 1.318 | 1.033 | **0.848** | 1.201 | 1.331 | 1.414 | 1.124 | 0.969 |
| | | | 192 | 1.195 | **0.884** | 1.024 | 1.319 | 1.054 | 1.222 | 1.094 | 0.909 | 1.287 | 1.061 | 1.109 | 1.058 | 0.941 |
| | | | 336 | 1.588 | 0.907 | 1.166 | 1.233 | **0.787** | 1.056 | 0.959 | 0.865 | 1.260 | 0.996 | 1.011 | 0.960 | 0.917 |
| | | MAE | 96 | 0.857 | 0.782 | 0.742 | 0.907 | 0.713 | 0.879 | 0.745 | 0.727 | 0.785 | 0.823 | 0.862 | 0.749 | **0.702** |
| | | | 192 | 0.783 | **0.701** | 0.749 | 0.891 | 0.721 | 0.855 | 0.737 | 0.757 | 0.831 | 0.726 | 0.755 | 0.722 | 0.737 |
| | | | 336 | 0.903 | 0.730 | 0.748 | 0.849 | **0.639** | 0.766 | 0.691 | 0.714 | 0.810 | 0.693 | 0.695 | 0.749 | 0.715 |

are no longer effective. Notably, MICN, which performs the worst under conventional data and settings, shows the best performance in our setting. This is understandable because Autoformer's core technology focuses on exploring the correlation within the data itself. This insight suggests that future efforts to tackle this problem should place greater emphasis on leveraging the intrinsic potential of the data. Additionally, we have compiled the training time for one epoch of all the baselines on the largest dataset, PEMS04_gap, and the smallest dataset, tfNSW, as shown in Table 7.

Considering that our dataset provides an extensive temporal range for training, we can theoretically extend the input step length considerably. We extended the 96 steps input from Table 3 to a maximum of 1440 steps for testing. As shown in Table 4, the performance improves significantly with the increase in input step length, further demonstrating the substantial potential of our dataset for deep exploration.

Table 4: Results of ablation study with 4 different input step lengths

| DLinear (Input Length) | | | 96 | | 192 | | 336 | | 720 | |
|---|---|---|---|---|---|---|---|---|---|---|
| Metrics | | | MSE | MAE | MSE | MAE | MSE | MAE | MSE | MAE |
| **PEMS03_gap** | 1-year gap | 96 | 1.500 | 0.933 | 1.455 | 0.912 | 1.462 | 0.906 | **1.392** | **0.887** |
| | | 192 | 1.542 | 0.945 | 1.147 | 0.910 | 1.457 | **0.906** | **1.445** | 0.906 |
| | | 336 | 1.531 | 0.935 | 1.447 | 0.907 | 1.462 | 0.906 | **1.439** | **0.904** |

## 5.5 RESULTS OF HOURLY AND DAILY FORECASTING

We believe that both the hourly and daily datasets are equally significant. Multi-scale, diverse datasets can provide the community with valuable references. We observe that the performance degrades progressively from the hourly to the daily to the gap datasets. Smaller time scales help reduce complexity and uncertainty, thereby improving prediction accuracy. Research has shown that clustering at different scales can enhance model performance Wang et al. (2024). Therefore, our aggregated version of the data will contribute new external features to the community.

Table 5: Comparison in hourly and daily datasets

| Methods | | | Mamba | | iTransformer | | DLinear | | Autoformer | |
|---|---|---|---|---|---|---|---|---|---|---|
| Metrics | | | MSE | MAE | MSE | MAE | MSE | MAE | MSE | MAE |
| | Hourly | 96 | 0.144 | 0.222 | 0.530 | 0.535 | 0.159 | 0.222 | 0.241 | 0.346 |
| | | 192 | 0.173 | 0.237 | 0.215 | 0.289 | 0.153 | 0.208 | 0.235 | 0.340 |
| | | 336 | 0.158 | 0.220 | 0.519 | 0.527 | 0.167 | 0.216 | 0.260 | 0.362 |
| **PEMS03_agg** | Daily | 96 | 0.754 | 0.503 | 0.606 | 0.419 | 0.602 | 0.426 | 0.771 | 0.537 |
| | | 192 | 0.968 | 0.604 | 0.781 | 0.500 | 0.794 | 0.509 | 0.897 | 0.577 |
| | | 336 | 1.210 | 0.706 | 0.967 | 0.579 | 0.984 | 0.584 | 1.058 | 0.630 |
| | Hourly | 96 | 0.137 | 0.244 | 0.240 | 0.339 | 0.161 | 0.245 | 0.178 | 0.295 |
| | | 192 | 0.132 | 0.239 | 0.260 | 0.361 | 0.142 | 0.223 | 0.175 | 0.288 |
| | | 336 | 0.121 | 0.216 | 0.226 | 0.413 | 0.145 | 0.226 | 0.197 | 0.316 |
| **PEMS04_agg** | Daily | 96 | 0.672 | 0.499 | 0.534 | 0.442 | 0.507 | 0.415 | 0.644 | 0.506 |
| | | 192 | 0.720 | 0.549 | 0.634 | 0.508 | 0.610 | 0.483 | 0.749 | 0.580 |
| | | 336 | 0.795 | 0.602 | 0.706 | 0.555 | 0.663 | 0.522 | 0.728 | 0.569 |
| | Hourly | 96 | 0.212 | 0.302 | 0.375 | 0.425 | 0.203 | 0.259 | 0.307 | 0.390 |
| | | 192 | 0.201 | 0.288 | 0.297 | 0.368 | 0.182 | 0.231 | 0.313 | 0.391 |
| | | 336 | 0.191 | 0.245 | 0.126 | 0.156 | 0.190 | 0.241 | 0.291 | 0.364 |
| **PEMS07_agg** | Daily | 96 | 1.719 | 0.736 | 1.426 | 0.613 | 1.414 | 0.606 | 1.703 | 0.762 |
| | | 192 | 2.005 | 0.842 | 1.772 | 0.730 | 1.756 | 0.720 | 1.903 | 0.804 |
| | | 336 | 2.290 | 0.949 | 2.078 | 0.819 | 2.051 | 0.813 | 2.171 | 0.884 |
| | Hourly | 96 | 0.245 | 0.287 | 0.363 | 0.379 | 0.253 | 0.272 | 0.305 | 0.377 |
| | | 192 | 0.269 | 0.292 | 0.341 | 0.354 | 0.254 | 0.259 | 0.340 | 0.401 |
| | | 336 | 0.283 | 0.298 | 0.369 | 0.369 | 0.281 | 0.272 | 0.452 | 0.468 |
| **PEMS08_agg** | Daily | 96 | 0.870 | 0.558 | 0.766 | 0.486 | 0.746 | 0.478 | 0.913 | 0.604 |
| | | 192 | 1.023 | 0.635 | 0.911 | 0.554 | 0.906 | 0.548 | 1.026 | 0.648 |
| | | 336 | 1.161 | 0.697 | 1.024 | 0.617 | 1.022 | 0.602 | 1.127 | 0.689 |

## 6 PROSPECTS AND CONSTRAINTS

**Prospects.** Our dataset spans the longest time period among existing datasets, and it is not only the largest spatially, but also evolving in growth. It can continue to update in line with the updates from the PeMS system in the future. It is specifically designed for various complex scenarios, such as those already mentioned, including extremely long forecasting with long gaps, and hourly and daily predictions. Additionally, zero-shot forecasting designed for evolving growth scenarios will also be included in Appendix.

**Constraints.** The limitations of our dataset are also quite evident. Due to the sheer size of our dataset, it requires more computational resources. However, with the maturation of large language models and foundational models, we believe its large volume will become an advantage, contributing more diverse data to the spatio-temporal large model community.

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

# A   APPENDIX

## A.1   DATASET

To support our setting of extremely long forecasting with gaps, we selected a subset installed in the early stages and have consistently collected new data up to the present, which is shown as follows:

Table 6: Overview of Gap, Hourly, and Daily Aggregated Data Based on Processed Raw Data

| Datasets(Gap/Hour/Day) | Time Period | Nodes |
|---|---|---|
| **PEMS03_gap&agg** | 03/2001 - 03/2024 | 151 |
| **PEMS04_gap&agg** | 06/2002 - 03/2024 | 822 |
| **PEMS05_gap&agg** | 03/2012 - 03/2024 | 103 |
| **PEMS06_gap&agg** | 12/2009 - 03/2024 | 130 |
| **PEMS07_gap&agg** | 06/2002 - 03/2024 | 3062 |
| **PEMS08_gap&agg** | 03/2001 - 03/2024 | 212 |
| **PEMS10_gap&agg** | 06/2007 - 03/2024 | 107 |
| **PEMS11_gap&agg** | 09/2002 - 03/2024 | 521 |
| **PEMS12_gap&agg** | 01/2002 - 03/2024 | 1543 |
| **tfNSW** | 01/2013 - 05/2024 | 27 |

## A.2   RESULTS

Table 7: Model Training Time Comparison. The training time for all baselines per epoch is measured in seconds.

| Baselines | Mamba | iTransformer | DLinear | Autoformer | Informer | FEDFormer | MICN | PatchTST | STGCN | ASTGCN | GWN | AGCRN | PDFormer |
|---|---|---|---|---|---|---|---|---|---|---|---|---|---|
| PEMS04_gap1 | 396.3 | 1310.2 | 237.9 | 1656.7 | 652.5 | 900.9 | 409.8 | 3649.4 | 2425.9 | 7921.9 | 4110.4 | 13369.1 | 8013.9 |
| tfNSW_gap1 | 13.7 | 44.1 | 7.4 | 148.2 | 73.3 | 380.1 | 101.2 | 55.3 | 37.3 | 228.8 | 50.4 | 658.6 | 47.9 |

Here we provided the results of District 5,6,10,11,12, as shown in Table 8 and Table 9:

Additionally, we provided the results of naive zero-shot forecasting, as shown in Table 10. The poor performance of this method indicates significant potential for improvement.

Table 8: Comparison in hourly and daily datasets in District 05,06,10,11,12. The symbol − indicates that the result is an outlier.

| Methods | | | Mamba | | iTransformer | | DLinear | | Autoformer | |
|---|---|---|---|---|---|---|---|---|---|---|
| Metrics | | | MSE | MAE | MSE | MAE | MSE | MAE | MSE | MAE |
| **PEMS05_agg** | Hourly | 96 | **0.121** | **0.205** | 0.226 | 0.324 | 0.148 | **0.236** | 0.155 | 0.272 |
| | | 192 | **0.118** | **0.197** | 0.214 | 0.312 | 0.132 | 0.216 | 0.159 | 0.268 |
| | | 336 | **0.115** | **0.194** | 0.220 | 0.318 | 0.134 | 0.217 | 0.167 | 0.274 |
| | Daily | 96 | 0.655 | 0.511 | 0.654 | 0.510 | **0.607** | **0.477** | 0.650 | 0.522 |
| | | 192 | 0.798 | 0.598 | 0.775 | 0.576 | 0.698 | 0.535 | **0.643** | **0.512** |
| | | 336 | 0.907 | 0.614 | 0.787 | 0.582 | **0.745** | **0.556** | 0.764 | 0.578 |
| **PEMS06_agg** | Hourly | 96 | **0.142** | **0.234** | 0.269 | 0.336 | 0.188 | 0.254 | 0.164 | 0.271 |
| | | 192 | **0.138** | **0.218** | 0.256 | 0.326 | 0.166 | 0.227 | 0.188 | 0.291 |
| | | 336 | **0.137** | **0.207** | 0.226 | 0.413 | 0.171 | 0.227 | 0.197 | 0.316 |
| | Daily | 96 | 0.516 | 0.419 | 0.414 | 0.389 | **0.405** | **0.340** | 0.518 | 0.437 |
| | | 192 | 0.642 | 0.485 | 0.543 | 0.468 | **0.510** | **0.395** | 0.580 | 0.460 |
| | | 336 | 0.734 | 0.519 | 0.675 | 0.492 | **0.596** | **0.437** | 0.658 | 0.481 |
| **PEMS10_agg** | Hourly | 96 | **0.213** | **0.256** | 0.391 | 0.413 | 0.272 | 0.309 | 0.260 | 0.346 |
| | | 192 | **0.205** | **0.250** | 0.387 | 0.412 | 0.246 | 0.277 | 0.380 | 0.417 |
| | | 336 | 0.211 | 0.255 | 0.394 | 0.413 | **0.258** | **0.281** | 0.329 | 0.386 |
| | Daily | 96 | 1.161 | 0.671 | **0.926** | **0.513** | 0.951 | 0.567 | 1.079 | 0.647 |
| | | 192 | 1.459 | 0.784 | 1.414 | 0.730 | **1.228** | **0.681** | 1.429 | 0.786 |
| | | 336 | 1.715 | 0.855 | 1.478 | 0.768 | **1.451** | **0.751** | 1.552 | 0.817 |
| **PEMS11_agg** | Hourly | 96 | - | 0.472 | - | 0.538 | - | **0.306** | - | 0.800 |
| | | 192 | - | 0.470 | - | 0.443 | - | **0.291** | - | 0.742 |
| | | 336 | - | 0.487 | - | 0.435 | - | **0.300** | - | 0.745 |
| | Daily | 96 | - | - | - | - | - | - | - | - |
| | | 192 | - | - | - | - | - | - | - | - |
| | | 336 | - | - | - | - | - | - | - | - |
| **PEMS12_agg** | Hourly | 96 | 0.145 | 0.237 | **0.083** | **0.157** | 0.174 | 0.242 | 0.188 | 0.287 |
| | | 192 | 0.142 | 0.226 | **0.091** | **0.159** | 0.154 | 0.216 | 0.196 | 0.289 |
| | | 336 | 0.146 | 0.217 | **0.104** | **0.170** | 0.162 | 0.219 | 0.209 | 0.298 |
| | Daily | 96 | 1.373 | 0.621 | 1.456 | 0.622 | **1.052** | **0.510** | 1.348 | 0.649 |
| | | 192 | 1.722 | 0.726 | 1.675 | 0.678 | **1.403** | **0.611** | 1.548 | 0.686 |
| | | 336 | 2.066 | 0.823 | 1.984 | 0.641 | **1.654** | **0.675** | 1.801 | 0.746 |

Table 9: Comparison in gap dataset in District 05,06,07,10,11,12. The bold text indicates the best.

| Methods | | | Mamba | | iTransformer | | DLinear | | Autoformer | |
|---|---|---|---|---|---|---|---|---|---|---|
| Metrics | | | MSE | MAE | MSE | MAE | MSE | MAE | MSE | MAE |
| PEMS05_gap | 1-year gap | 96 | 2.079 | 1.209 | 1.945 | 1.164 | 1.291 | 0.916 | **1.065** | **0.796** |
| | | 192 | 2.132 | 1.256 | 1.984 | 1.185 | 1.750 | 1.099 | **1.063** | **0.809** |
| | | 336 | 2.377 | 1.340 | 2.067 | 1.234 | 1.894 | 1.144 | **1.135** | **0.827** |
| | 1.5-year gap | 96 | 1.852 | 1.122 | 1.879 | 1.078 | 1.683 | 1.054 | **1.060** | **0.785** |
| | | 192 | 1.929 | 1.182 | 1.593 | 1.032 | 1.633 | 1.045 | **0.912** | **0.712** |
| | | 336 | 2.370 | 1.313 | 2.214 | 1.071 | 0.794 | 1.184 | **1.345** | **0.882** |
| | 2-year gap | 96 | 1.868 | 1.106 | 1.580 | 0.969 | 1.602 | 1.018 | **0.828** | **0.672** |
| | | 192 | 2.219 | 1.274 | 1.481 | 0.958 | 1.589 | 1.027 | **1.018** | **0.772** |
| | | 336 | 2.695 | 1.212 | 2.207 | 1.201 | 1.922 | 1.139 | **1.186** | **0.839** |
| PEMS06_gap | 1-year gap | 96 | 1.806 | 1.066 | 0.875 | 0.692 | 1.173 | 0.837 | **1.216** | **0.859** |
| | | 192 | 1.928 | 1.112 | 1.227 | 0.848 | 1.410 | 0.942 | **0.961** | **0.751** |
| | | 336 | 2.181 | 1.212 | 1.594 | 1.003 | 1.501 | 0.976 | **0.992** | **0.769** |
| | 1.5-year gap | 96 | 1.549 | 0.997 | 1.331 | 0.891 | 1.484 | 0.963 | **0.885** | **0.710** |
| | | 192 | 1.746 | 1.054 | 1.077 | 0.778 | 1.353 | 0.920 | **1.010** | **0.768** |
| | | 336 | 1.605 | 1.018 | 1.500 | 0.961 | 1.587 | 1.011 | **0.955** | **0.739** |
| | 2-year gap | 96 | 1.226 | 0.851 | 1.864 | 1.106 | 1.691 | 1.033 | **1.013** | **0.768** |
| | | 192 | 0.949 | 0.720 | 1.343 | 0.879 | 1.259 | 0.858 | **0.853** | **0.691** |
| | | 336 | **0.945** | **0.710** | 1.550 | 0.970 | 1.415 | 0.934 | 0.955 | 0.739 |
| PEMS07_gap | 1-year gap | 96 | 1.719 | 1.006 | 1.756 | 1.024 | 1.680 | 1.024 | **1.215** | **0.867** |
| | | 192 | 1.637 | 0.996 | 1.816 | 1.061 | 1.776 | 1.043 | **1.202** | **0.852** |
| | | 336 | 1.637 | 0.991 | 1.784 | 1.028 | 1.774 | 1.034 | **1.098** | **0.774** |
| | 1.5-year gap | 96 | 1.571 | 0.972 | 1.705 | 1.016 | 1.585 | 0.987 | **1.184** | **0.844** |
| | | 192 | 1.578 | 0.982 | 1.657 | 1.004 | 1.624 | 0.996 | **1.088** | **0.800** |
| | | 336 | 1.818 | 1.127 | 1.712 | 1.028 | 1.668 | 1.036 | **0.991** | **0.729** |
| | 2-year gap | 96 | 5.411 | 1.319 | 2.055 | 1.149 | 1.746 | 1.048 | **1.099** | **0.802** |
| | | 192 | 12.620 | 1.370 | 2.006 | 1.128 | 1.705 | 1.002 | **1.284** | **0.875** |
| | | 336 | 9.614 | 1.378 | 1.812 | 1.057 | 1.605 | 0.968 | **0.972** | **0.734** |
| PEMS10_gap | 1-year gap | 96 | 2.310 | 1.233 | **0.882** | **0.698** | 1.466 | 0.963 | 1.208 | 0.865 |
| | | 192 | 2.878 | 1.396 | 1.606 | 0.993 | 1.834 | 1.106 | **1.152** | **0.860** |
| | | 336 | 2.584 | 1.327 | 1.413 | **0.825** | 1.887 | 1.121 | **1.130** | 0.830 |
| | 1.5-year gap | 96 | 2.181 | 1.206 | 1.812 | 1.052 | 1.765 | 1.057 | **1.368** | **0.913** |
| | | 192 | 2.525 | 1.318 | 1.645 | 0.986 | 1.791 | 1.069 | **1.151** | **0.828** |
| | | 336 | 2.488 | 1.310 | 2.051 | 1.134 | 1.976 | 1.139 | **1.111** | **0.809** |
| | 2-year gap | 96 | **1.165** | **0.822** | 2.772 | 1.383 | 1.977 | 1.119 | 1.188 | 0.845 |
| | | 192 | 1.082 | 0.792 | 1.974 | 1.084 | 1.490 | 0.932 | **0.971** | **0.751** |
| | | 336 | **1.021** | **0.764** | 2.157 | 1.168 | 1.619 | 0.989 | 1.096 | 0.809 |
| PEMS11_gap | 1-year gap | 96 | 5.199 | 0.891 | 5.417 | 0.997 | 5.276 | 0.936 | **4.930** | **0.854** |
| | | 192 | 5.300 | 0.920 | 5.504 | 1.029 | 5.393 | 0.974 | **5.114** | **0.867** |
| | | 336 | 5.251 | 0.901 | 5.410 | 0.985 | 5.364 | 0.955 | **4.830** | **0.849** |
| | 1.5-year gap | 96 | 5.871 | 0.968 | 6.045 | 1.296 | 5.914 | 1.224 | **5.691** | **0.878** |
| | | 192 | 5.968 | 1.012 | 6.121 | 1.314 | 5.993 | 1.229 | **5.792** | **0.884** |
| | | 336 | 6.167 | 1.074 | 6.214 | 1.326 | 6.025 | 1.299 | **5.947** | **0.967** |
| | 2-year gap | 96 | 5.914 | 0.996 | 6.136 | 1.318 | 5.945 | 1.243 | **5.761** | **0.978** |
| | | 192 | 6.541 | 1.043 | 6.213 | 1.327 | 6.014 | 1.289 | **6.245** | **1.001** |
| | | 336 | 6.541 | 1.086 | 6.221 | 1.332 | 6.024 | 1.300 | **6.268** | **1.024** |
| PEMS12_gap | 1-year gap | 96 | 1.751 | 1.025 | 1.624 | 1.002 | 1.611 | 1.005 | **1.060** | **0.789** |
| | | 192 | 1.726 | 1.024 | 1.424 | 0.929 | 1.537 | 0.972 | **1.150** | **0.840** |
| | | 336 | 1.751 | 1.029 | 1.672 | 1.015 | 1.683 | 1.017 | **0.889** | **0.719** |
| | 1.5-year gap | 96 | 1.554 | 0.967 | 1.479 | 0.921 | 1.468 | 0.910 | **0.954** | **0.875** |
| | | 192 | 1.401 | 0.898 | 1.314 | 0.867 | 1.301 | 0.862 | **0.943** | **0.869** |
| | | 336 | 1.417 | 0.906 | 1.322 | 0.869 | 1.298 | 0.859 | **0.921** | **0.846** |
| | 2-year gap | 96 | 0.956 | 0.704 | 0.876 | 0.671 | 0.872 | 0.664 | **0.846** | **0.659** |
| | | 192 | 0.846 | 0.656 | 0.813 | 0.653 | 0.806 | 0.649 | **0.785** | **0.628** |
| | | 336 | 0.814 | 0.628 | 0.789 | 0.631 | 0.776 | 0.624 | **0.754** | **0.617** |

Table 10: Results of zero shot forecasting in PEMS03. In this study, we utilized the inputs from 151 existing nodes in PEMS03 to predict 52 new nodes, resulting in a test set comprising a total of 203 nodes.

| Datasets | | 96 | | 192 | | 336 | |
|---|---|---|---|---|---|---|---|
| Metrics | | MSE | MAE | MSE | MAE | MSE | MAE |
| | 1-year gap | 4.658 | 3.465 | 4.945 | 3.648 | 5.198 | 3.892 |
| PEMS03_gap | 1.5-year gap | 4.891 | 3.657 | 5.124 | 3.842 | 5.263 | 3.996 |
| | 2-year gap | 4.547 | 3.410 | 4.895 | 3.539 | 5.103 | 3.758 |

