# OpenReview forum: "XXLTraffic: Expanding and Extremely Long Traffic forecasting beyond test adaptation"
_ICLR.cc/2025/Conference — ICLR 2025 Conference Withdrawn Submission_

### Official Review · Reviewer_KwQa · 2024-10-26

**Soundness:** 4
**Presentation:** 3
**Contribution:** 4
**Rating:** 8
**Confidence:** 4

**Summary:**

This paper present an XXLTraffic traffic dataset spanning up to 23 years in different countries, which can be useful for the other researchers. They have also implemented the datasets with many advanced approaches to confirm the range and different experiments directions.

**Strengths:**

1. This paper introduces an XXLTraffic dataset that spans up to 23 years from different regions, which means the dataset include sufficient information for the traffic feature.
2. This paper presents many potential conditions that the dataset could be applied to such as short-term traffic prediction, long-term multivariate traffic prediction and extremely long term prediction with gaps. All of these domains are worthwhile problem to focus.

**Weaknesses:**

1. The dataset is valuable, but performing extremely long-term predictions is challenging, as traffic patterns can change significantly over the years, and even the road infrastructure itself may be reshaped over a 23-year span. Such factors could potentially reduce the impact of the dataset's contribution, because many open datasets are enough for normal short and long term prediction.
2. Some typos, e.g., ‘beyongd’  should be ‘beyond’ around line 130;

**Questions:**

How is the gap dataset collected, given that "gap" typically refers to missing data? Can this dataset be directly used to address traditional missing data scenarios?

---

> ### Author Response · Authors · 2024-11-25
>
> We appreciate the reviewer's acknowledgment of the dataset’s impact and the careful review for our manuscript.
>
> **Q1**: Road infrastructure may influence the long time span dataset.
> > **The dataset is valuable, but performing extremely long-term predictions is challenging, as traffic patterns can change significantly over the years, and even the road infrastructure itself may be reshaped over a 23-year span. Such factors could potentially reduce the impact of the dataset's contribution, because many open datasets are enough for normal short and long term prediction.**
>
> We appreciate your feedback and understand your concern regarding the factor of traffic patterns changing over a 23-year span. Therefore, we set the gap years to several years, with the gap length referencing the time it takes for a highway to be planned and finally completed. Predicting the traffic volume at those locations several years in advance is of great significance. As you can see from Table 1 in the paper, the performance of all baselines is relatively poor, indicating the considerable difficulty of accurately predicting this kind of setting. Thus, our dataset spanning up to 23 years can provide more samples for training and offer more potential for modeling temporal features in the models.
>
> **Q2**: Some typos.
> > **Some typos, e.g., ‘beyongd’ should be ‘beyond’ around line 130;**
>
> Thank you for your feedback. We will carefully review the entire manuscript to correct any grammatical errors and typos. These are some error corrections from our self-review as follows (the full table could be seen in the author global rebuttal):
>
> (1) In Section 1.1, "Our dataset are evolving and longer than existing datasets." should be "Our dataset is...".
>
> (2) In Section 4.1, "tfNSW is an open-source data platform provided by Transport for NSW" should be "The tfNSW is...." because "tfNSW" is the first word of the sentence.
>
> (3) In Section. 5.4, "SOTA" should be "the state of the art" since it is being used for the first time.
>
> Additionally, Based on Reviewer ytp5’s suggestion, we have added five additional baselines shown in the table:
>
> ||||**STGCN**||**ASTGCN**||**GWN**||**AGCRN**||**PDFormer**||
> |-|-|-|-|-|-|-|-|-|-|-|-|-|
> |**Methods**|**Metrics**||**MSE**|**MAE**|**MSE**|**MAE**|**MSE**|**MAE**|**MSE**|**MAE**|**MSE**|**MAE**|
> |||96|0.556|0.536|0.765|0.618|0.676|0.574|0.596|0.562|0.621|0.576|
> ||**1-yeargap**|192|0.562|0.539|0.764|0.626|0.581|0.546|0.565|0.543|0.545|0.536|
> |||336|0.561|0.538|0.717|0.598|0.582|0.543|0.580|0.552|0.574|0.548|
> |||96|1.256|0.884|1.441|0.959|1.250|0.875|1.168|0.866|1.256|0.899|
> |**PEMS03\_gap**|**1.5-yeargap**|192|1.187|0.872|1.395|0.947|1.194|0.856|1.122|0.843|1.117|0.842|
> |||336|1.167|0.853|1.273|0.877|1.1015|0.832|1.176|0.862|1.182|0.871|
> |||96|2.059|1.176|2.236|1.234|1.987|1.148|1.950|1.150|2.068|1.189|
> ||**2-yeargap**|192|2.055|1.184|2.180|1.213|1.877|1.117|1.746|1.070|1.897|1.131|
> |||336|2.022|1.169|1.889|1.117|1.847|1.107|1.930|1.140|1.982|1.161|
>
> Based on Reviewer 6qSU’s suggestion, we have provided statistics on training times for the largest and smallest datasets shown in the following table, demonstrating that our dataset and the settings we propose can be easily trained on a standard single V100 GPU.
>
> ||||**Mamba**||**iTransformer**||**DLinear**||**Autoformer**||**Informer**||**FEDFormer**||**MICN**||**PatchTST**||
> |-|-|-|-|-|-|-|-|-|-|-|-|-|-|-|-|-|-|-|
> |**Methods**|**Metrics**||**MSE**|**MAE**|**MSE**|**MAE**|**MSE**|**MAE**|**MSE**|**MAE**|**MSE**|**MAE**|**MSE**|**MAE**|**MSE**|**MAE**|**MSE**|**MAE**|
> |||96|1.480|0.845|1.236|0.822|1.166|0.748|1.320|0.879|1.354|0.777|1.377|0.895|1.219|0.778|0.946|0.778|
> ||**1-yeargap**|192|1.543|0.867|1.088|0.733|1.184|0.758|1.599|0.961|1.229|0.757|1.312|0.890|1.262|0.799|1.026|0.821|
> |||336|1.459|0.847|1.099|0.739|1.196|0.763|1.477|0.934|1.242|0.776|1.219|0.838|1.366|0.833|0.949|0.774|
> |||96|1.745|0.944|1.355|0.877|1.372|0.846|1.658|1.016|1.615|0.879|1.440|0.919|1.204|0.778|1.025|0.807|
> |**tfNSW\_gap**|**1.5-yeargap**|192|1.784|0.968|1.260|0.818|1.367|0.846|1.666|0.997|1.484|0.854|1.452|0.932|1.251|0.789|1.058|0.823|
> |||336|1.628|0.939|1.284|0.835|1.385|0.854|1.515|0.945|1.500|0.883|1.346|0.871|1.203|0.770|0.995|0.788|
> |||96|1.364|0.857|1.019|0.782|1.008|0.742|1.319|0.907|1.101|0.713|1.318|0.879|1.033|0.745|0.848|0.727|
> ||**2-yeargap**|192|1.195|0.783|0.884|0.701|1.024|0.749|1.319|0.891|1.054|0.721|1.222|0.855|1.094|0.737|0.909|0.757|
> |||336|1.588|0.903|0.907|0.730|1.166|0.748|1.233|0.849|1.787|0.639|1.056|0.766|0.959|0.691|0.865|0.714|

---

> > ### Comment · Reviewer_KwQa · 2024-11-25
> >
> > Thanks for the authors reply. They have answered most of my questions. I will maintain my score.

---

### Official Review · Reviewer_YRBw · 2024-11-02

**Soundness:** 2
**Presentation:** 3
**Contribution:** 2
**Rating:** 5
**Confidence:** 5

**Summary:**

This paper proposed large-scale datasets, named XXLTrafic, to address limitations in existing traffic forecasting datasets. It provides extensive, real-world datasets with long timespans to support research in extremely long forecasting and adaptive model development for evolving traffic patterns.

**Strengths:**

1.	The proposed XXLTraffic is the largest publicly available traffic dataset so far. The PEMS part spans up to 23 years and the NSW part also has around 11 years’ data.

2.	The authors evaluated the performance of existing baselines on the proposed datasets with various settings. The overall results verify the importance of introducing large-scale datasets with extremely long time spans.

**Weaknesses:**

1.	The most significant weakness is that this paper has no novelty. Although this conference has a dataset and benchmarks area, I do not think, for a top-tier conference, it is enough to propose a large-scale dataset without any model contribution. Given the fact that there are existing large-scale datasets available with just the number of nodes or time spans fewer than XXTraffic, my concern becomes more intense.

2.	Some motivations remain unclear. In the introduction section, the authors mentioned ‘beyond test adaptation’. However, it is not easy to understand how this relates to the proposed datasets. Is the concept of ‘beyond test adaptation’ better or more realistic than test time adaptation?

3.	The setting of Extremely Long Prediction with Gaps is a bit strange to me. Here, the authors used an example of highway route planning prediction to justify the necessity of the gap. It is possible that we need to stimulate the traffic flows of uninstalled sensors to facilitate road planning. Thus, the optimal setting should be using existing sensors to predict traffic flows of uninstalled sensors, plus the gap. However, it seems the authors still used historical signals of uninstalled sensors in this setting. If we already know their historical readings, we do not have to plan anything.

**Questions:**

The authors can address the above concerns.

---

> ### Author Response · Authors · 2024-11-25
>
> We sincerely thank the reviewer for taking the time to provide valuable suggestions and detailed comments. We are pleasant to hear that the reviewer'acknowledges of importance of introducing large-scale datasets with extremely long time spans.
>
> **Q1**: Lack of model contribution
> > **The most significant weakness is that this paper has no novelty. Although this conference has a dataset and benchmarks area, I do not think, for a top-tier conference, it is enough to propose a large-scale dataset without any model contribution. Given the fact that there are existing large-scale datasets available with just the number of nodes or time spans fewer than XXTraffic, my concern becomes more intense.**
>
> We appreciate your feedback and understand your concern regarding the novelty of our submission. However, we would like to point out that many esteemed dataset contributions in the field have focused primarily on the datasets themselves and the associated baseline experiments, without necessarily introducing a new model. Examples include the Large-ST dataset from NIPS 2023 and the DL-Traffic dataset, which was a runner-up for the best paper at CIKM 2021. Neither of these proposed their own models, yet they are recognized for their significant contributions to the research community.
>
> We firmly believe that the data processing, experimental setup, and the extensive baseline comparisons we have presented hold practical research value. In response to the concerns raised by Reviewer ytp5 and Reviewer 6qSU, we have added five additional baselines and included statistics on training times for the largest and smallest datasets, demonstrating that our dataset and the settings we propose can be easily trained on a standard single V100 GPU.
>
> We maintain that our work provides a substantial contribution to the field by offering a unique dataset that enables long-term traffic forecasting with gaps, a challenging domain that has not been sufficiently explored. The practical implications of our dataset and experimental framework are significant, and we are confident that they will facilitate further research and innovation in traffic forecasting.
>
> [1] Liu X, Xia Y, Liang Y, et al. Largest: A benchmark dataset for large-scale traffic forecasting[J]. Advances in Neural Information Processing Systems, 2024, 36.
>
> [2] Jiang R, Yin D, Wang Z, et al. Dl-traff: Survey and benchmark of deep learning models for urban traffic prediction[C]//Proceedings of the 30th ACM international conference on information & knowledge management. 2021: 4515-4525.
>
> **Q2**: Unclear destription of some motivation
> > **Some motivations remain unclear. In the introduction section, the authors mentioned ‘beyond test adaptation’. However, it is not easy to understand how this relates to the proposed datasets. Is the concept of ‘beyond test adaptation’ better or more realistic than test time adaptation?**
>
> Thank you for your suggestion. We have clarified the concepts of 'beyond test adaptation' and 'test time adaptation' in the introduction section to distinguish our approach from test time adaptation. Our aim is to provide a dataset with an extended time span that can accommodate various durations of gap settings, such as 1 year, 5 years, and in this work, we have recommended 1, 1.5, and 2 years for baseline comparison experiments. Under different gap settings, we can consider them as different distributions. We will expand Figure 1 in subsequent versions to include gap settings and clarify the concept more clearly.

---

> > ### Author Response · Authors · 2024-11-25
> >
> > **Q3**: Unclear destription of some motivation
> > > **The setting of Extremely Long Prediction with Gaps is a bit strange to me. Here, the authors used an example of highway route planning prediction to justify the necessity of the gap. It is possible that we need to stimulate the traffic flows of uninstalled sensors to facilitate road planning. Thus, the optimal setting should be using existing sensors to predict traffic flows of uninstalled sensors, plus the gap. However, it seems the authors still used historical signals of uninstalled sensors in this setting. If we already know their historical readings, we do not have to plan anything.**
> >
> > Our experiments with gap settings are based on the historical datasets of installed sensors, as only an extremely long time span dataset can support such configurations. Utilizing historical data from existing sensors to predict future traffic flows during gap years is a relatively simpler scenario. Our baseline comparison experiments reveal that even under this simplified setting, accurate prediction is quite challenging. If we use data from installed sensors to predict traffic flows for uninstalled sensors shown in Table 9 (Appendix part), the results would be several times worse compared to the settings in Table 1 of our work. The extremely time span dataset we propose offers more potential to address the aforementioned challenges, both by enriching the data samples on the temporal dimension and by providing more possibilities for temporal feature extraction, as demonstrated in Table 4 (in the manuscript) where lengthening the data input enhances forecasting performance.
> >
> >
> > Additionally, Based on Reviewer ytp5’s suggestion, we have added five additional baselines shown in Table 1 shown in global rebuttal. And based on Reviewer 6qSU’s suggestion, we have provided statistics on training times for the largest and smallest datasets shown in Table 2 in global rebuttal, demonstrating that our dataset and the settings we propose can be easily trained on a standard single V100 GPU.

---

> > > ### Comment · Reviewer_YRBw · 2024-11-26
> > > **Thanks for your response**
> > >
> > > The authors clarified the motivations and some ambiguous of this paper, which is helpful. However, the major concern of novelty still cannot be (fully) addressed.
> > >
> > > Large-ST was published on the NeurIPS dataset and benchmark track and DL-Traffic was published on the CIKM resource track. These tracks specifically aim at dataset papers with their unique review criteria and submission channels. The proceedings also indicate the track explicitly. However, ICLR just mentions the datasets and benchmarks as a submittable topic. All papers are submitted to the same channel. Therefore, it is not enough to use Large-ST and DL-Traffic to justify the contributions for an ICLR submission.
> > >
> > > This is the major consideration for me to give the current score. If the paper is rejected, I recommend the authors to submit it to a better venue.
> > >
> > > Thanks.

---

> > > > ### Author Response · Authors · 2024-11-28
> > > > **Thanks for your review**
> > > >
> > > > Dear reviewer YRBw, we thank you again for your constructive review and comments. We hope we have addressed all your concerns. If you believe our comments and revisions have satisfied your concerns, would you please reconsider raising your score?

---

### Official Review · Reviewer_ytp5 · 2024-11-04

**Soundness:** 1
**Presentation:** 3
**Contribution:** 2
**Rating:** 5
**Confidence:** 4

**Summary:**

This paper presents a traffic dataset containing extensive data records from multiple regions over a long-term period, aimed at supporting research in ultra-long-term traffic prediction.

**Strengths:**

(1) A new traffic dataset is proposed, comprising data from numerous regions across California and New South Wales over an extended time period.

(2) The temporal distribution evolution of selected data was visualized, revealing the evolutionary characteristics of the data.

(3) Tests were conducted on relevant prediction tasks, including scenarios with time gaps and varying input lengths.

**Weaknesses:**

(1) Compared to existing works like LargeST, this work's contribution is insufficient as it merely expands the temporal and spatial scope of data collection. These data can be obtained through the open-source PEMS system in the same way.

(2) Although the number of regions and time span are substantial, the vast majority of data is limited to California. Including data from more cities and countries might have been a better approach.

(3) Most baselines in the experiments are specifically designed for time series prediction, lacking models specifically designed for traffic prediction/spatiotemporal prediction, such as PDFormer[1], ASTGCN[2], STGCN[3], AGCRN[4], GWN[5], etc. It is important to explore the performance of these types of baselines on traffic datasets.

[1] PDFormer: Propagation Delay-Aware Dynamic Long-Range Transformer for Traffic Flow Prediction. AAAI 2023.

[2] Attention Based Spatial-Temporal Graph Convolutional Networks for Traffic Flow Forecasting. AAAI 2019.

[3] Spatio-Temporal Graph Convolutional Networks: A Deep Learning Framework for Traffic Forecasting. IJCAI 2018.

[4] Adaptive Graph Convolutional Recurrent Network for Traffic Forecasting. NeurIPS 2020.

[5] Graph WaveNet for Deep Spatial-Temporal Graph Modeling. IJCAI 2019.

**Questions:**

Please refer to the Weaknesses.

---

> ### Author Response · Authors · 2024-11-25
>
> We thank the reviewer for taking the time to assess our manuscript and offering valuable suggestions.
>
> **Q1**: Insufficient contribution
> > **Compared to existing works like LargeST, this work's contribution is insufficient as it merely expands the temporal and spatial scope of data collection. These data can be obtained through the open-source PEMS system in the same way.**
>
> We acknowledge your concern regarding the comparison with existing works like LargeST and the perceived insufficiency in our contribution. It is true that our dataset, like LargeST, is derived from the open-source PEMS system. However, our primary contribution lies in the unprecedented expansion of the temporal dimension, which is a significant departure from datasets that focus more on spatial expansion, such as LargeST. Our dataset's unique value proposition is its ability to support extremely long-term traffic forecasting with gaps, a capability not commonly found in other datasets. This feature is particularly relevant for practical applications such as long-range urban planning and infrastructure development, which require traffic volume assessments years in advance. By providing data that spans over two decades, we enable researchers and planners to assess traffic patterns and trends over an extended period, which is crucial for strategic planning and decision-making. Furthermore, we have enriched our dataset with various baselines to highlight the challenges of the traffic forecasting domain, especially with our gap setting. This approach not only demonstrates the dataset's utility but also contributes to the field by providing a benchmark for evaluating the performance of different models in long-term forecasting scenarios. In summary, while our dataset may not exceed LargeST in terms of spatial dimensions, its extensive temporal coverage and the inclusion of gap settings offer a novel and valuable resource for traffic forecasting research and applications.
>
> **Q2**: Regional limitation of the dataset
> > **Although the number of regions and time span are substantial, the vast majority of data is limited to California. Including data from more cities and countries might have been a better approach.**
>
> Yes, we agree with your perspective that focusing solely on traffic data from California is limiting. Datasets from different countries and regions would indeed enrich the understanding of traffic conditions in various cities. Therefore, our dataset also includes a 13-year traffic dataset from New South Wales, Australia, which is complemented by the same set of baselines to ensure comparability and analysis across different geographical contexts.
>
> **Q3**: Additional traffic forecasting baselines
> > **Most baselines in the experiments are specifically designed for time series prediction, lacking models specifically designed for traffic prediction/spatiotemporal prediction, such as PDFormer[1], ASTGCN[2], STGCN[3], AGCRN[4], GWN[5], etc. It is important to explore the performance of these types of baselines on traffic datasets.**
>
> We agree with your suggestion. The reason we chose baselines based on the long-term forecasting domain for our experiments is that our experimental setup is more closely aligned with multivariate long-term forecasting. Since the introduction of Informer, the development of baselines in this field has been particularly vibrant, and we have benefited from the widely available open-source repositories that facilitate easy migration and reproduction. However, we acknowledge that it is also necessary to consider baselines from the traffic forecasting domain (12-step forecasting).

---

> ### Author Response · Authors · 2024-11-25
>
> **Q3**: Additional traffic forecasting baselines
>
> Therefore, based on the five baselines you provided, We have observed that most traffic forecasting baselines indeed perform worse compared to those in long-term forecasting, and their performance deteriorates as the gap increases, as shown in the following table (We also provided a traing time statistics icluding the aforementioned baselines and your suggested baselines in Table 2 of global rebuttal):
>
> ||||**STGCN**||**ASTGCN**||**GWN**||**AGCRN**||**PDFormer**||
> |-|-|-|-|-|-|-|-|-|-|-|-|-|
> |**Methods**|**Metrics**||**MSE**|**MAE**|**MSE**|**MAE**|**MSE**|**MAE**|**MSE**|**MAE**|**MSE**|**MAE**|
> |||96|0.556|0.536|0.765|0.618|0.676|0.574|0.596|0.562|0.621|0.576|
> ||**1-yeargap**|192|0.562|0.539|0.764|0.626|0.581|0.546|0.565|0.543|0.545|0.536|
> |||336|0.561|0.538|0.717|0.598|0.582|0.543|0.580|0.552|0.574|0.548|
> |||96|1.256|0.884|1.441|0.959|1.250|0.875|1.168|0.866|1.256|0.899|
> |**PEMS03\_gap**|**1.5-yeargap**|192|1.187|0.872|1.395|0.947|1.194|0.856|1.122|0.843|1.117|0.842|
> |||336|1.167|0.853|1.273|0.877|1.1015|0.832|1.176|0.862|1.182|0.871|
> |||96|2.059|1.176|2.236|1.234|1.987|1.148|1.950|1.150|2.068|1.189|
> ||**2-yeargap**|192|2.055|1.184|2.180|1.213|1.877|1.117|1.746|1.070|1.897|1.131|
> |||336|2.022|1.169|1.889|1.117|1.847|1.107|1.930|1.140|1.982|1.161|
> |||96|1.233|0.904|1.311|0.929|1.310|0.87|1.290|0.865|1.101|0.851|
> ||**1-yeargap**|192|1.398|1.025|1.443|1.023|1.255|0.879|1.318|0.907|1.078|0.816|
> |||336|1.505|1.104|1.325|0.938|1.292|0.865|1.302|0.899|1.085|0.821|
> |||96|1.369|1.082|1.350|0.937|1.684|1.063|1.222|0.961|1.097|0.849|
> |**PEMS04\_gap**|**1.5-yeargap**|192|1.642|1.035|1.549|1.075|1.501|0.988|1.346|0.989|1.214|0.903|
> |||336|1.368|0.862|1.199|0.832|1.584|0.997|0.231|0.963|1.130|0.854|
> |||96|1.653|1.074|1.247|0.894|1.669|1.057|1.236|0.901|1.099|0.861
> ||**2-yeargap**|192|1.545|1.023|1.356|0.948|1.554|1.041|1.269|0.915|1.159|0.891|
> |||336|1.556|1.011|1.174|0.842|1.128|0.869|1.314|0.947|1.032|0.816|
> |||96|2.531|1.325|3.119|1.404|2.185|1.280|2.158|1.220|1.921|1.148|
> ||**1-yeargap**|192|2.560|1.337|2.405|1.276|2.223|1.311|2.144|1.213|2.248|1.260|
> |||336|2.482|1.317|2.086|1.181|2.228|1.312|2.036|1.185|1.994|1.179|
> |||96|1.772|1.084|2.370|1.192|1.309|0.896|1.166|0.808|1.664|1.055|
> |**PEMS08\_gap**|**1.5-yeargap**|192|1.522|0.979|1.730|1.043|1.074|0.805|1.144|0.810|1.518|1.012|
> |||336|1.495|0.966|1.444|0.948|1.533|0.999|1.078|0.784|1.049|0.801|
> |||96|1.336|0.868|1.962|1.069|1.789|1.017|1.393|0.856|1.071|0.792|
> ||**2-yeargap**|192|1.218|0.819|1.292|0.795|1.007|0.761|1.229|0.813|1.171|0.826|
> |||336|1.181|0.806|1.227|0.842|1.181|0.813|1.246|0.814|1.003|0.768|
> |||96|1.176|0.776|1.404|0.874|1.451|0.871|1.283|0.824|1.070|0.756|
> ||**1-yeargap**|192|1.495|0.880|1.547|0.889|1.580|0.904|1.267|0.828|1.019|0.753|
> |||336|1.517|0.901|1.617|0.910|1.534|0.892|1.048|0.786|0.822|0.710|
> |||96|1.220|0.796|1.286|0.813|1.598|0.940|1.249|0.809|1.153|0.790|
> |**TfNSW\_gap**|**1.5-yeargap**|192|1.624|0.934|1.615|0.902|1.620|0.950|1.228|0.808|1.095|0.792|
> |||336|1.621|0.945|1.621|0.900|1.220|0.814|1.163|0.811|1.049|0.775|
> |||96|1.201|0.785|1.331|0.823|1.414|0.862|1.124|0.749|0.969|0.702|
> ||**2-yeargap**|192|1.287|0.831|1.061|0.726|1.109|0.755|1.058|0.722|0.941|0.737|
> |||336|1.260|0.810|0.996|0.693|1.011|0.695|0.960|0.749|0.917|0.715|

---

> > ### Comment · Reviewer_ytp5 · 2024-11-26
> >
> > Thank you for the authors' response, which addressed some of my concerns. The authors supplemented extensive experimental results of spatiotemporal baselines on the datasets, making the work more complete, and I will increase my score to acknowledge this improvement. However, I still believe that the main contribution of this paper lies in expanding the temporal range of data collection and releasing some new datasets. Compared to technical papers at ICLR, the contribution seems insufficient. As a dataset paper, I think it might be more suitable for publication in dedicated tracks (such as NeurIPS's dataset and benchmark track).

---

### Official Review · Reviewer_6qSU · 2024-11-04

**Soundness:** 3
**Presentation:** 2
**Contribution:** 3
**Rating:** 5
**Confidence:** 4

**Summary:**

The authors present XXLTraffic, the longest available public traffic dataset with the longest timespan collected from Los Angeles, USA, and New South Wales, Australia, curated to support research in extremely long forecasting beyond test adaptation. The benchmark includes both typical time-series forecasting settings with hourly and daily aggregated data and novel configurations that introduce gaps and down-sample the training size to better simulate practical constraints.

**Strengths:**

1. A brand-new dataset is proposed, which is the largest available public traffic dataset in extremely long traffic forecasting.

2. Numerous comparative experiments validate the differences in performance of different models on this dataset, providing a good guide for peers to understand the latest technological developments.

**Weaknesses:**

1. The maximum number of nodes in the proposed dataset is less than the Large-ST, which seems to need improvement, after all, the real road network is very large.

2. I think it is necessary for the authors to explain in more depth the motivation for constructing such a long-term dataset, because as transportation infrastructure advances and human travel modes change, traffic data that is too ancient is not always helpful enough to understand future transportation patterns.

3.  Due to the sheer size of the proposed dataset, it requires more computational resources. Therefore, whether this dataset can be popularized and whether peers can easily adopt it in their own research is also a concern.

**Questions:**

See Weaknesses.

---

> ### Author Response · Authors · 2024-11-25
>
> We sincerely thank for the constructive comments and suggestions. We are encouraged that the reviewer acknowledges that our work provide a good guide for peers to understand the latest technological developments.
>
> **Q1**: Spatial size of the dataset
> > **The maximum number of nodes in the proposed dataset is less than the Large-ST, which seems to need improvement, after all, the real road network is very large.**
>
> Thank you for your review. We appreciate the feedback provided. In the hyperlink in our manuscript, we have made available the raw dataset which contains a substantial number of nodes. However, our focus is on conducting extremely long-term forecasting (i.e., our gap setting, such as gaps of 1 year, 1.5 years, and 2 years) based on nodes that span over twenty years. However, most of these nodes do not have data spanning over 20 years, which is essential for our gap setting. After conducting data analysis and preprocessing, we have established that each district contains between 100 and 800 nodes. This is why our spatial node count does not reach the substantial numbers seen in the Large-ST dataset.
>
> Despite the relatively lower number of nodes per district in our dataset, we offer datasets from various geographical areas, including 9 districts from the PeMS dataset and 1 traffic dataset from the state of New South Wales, Australia. This diverse regional coverage allows for a comprehensive exploration of extremely long-term traffic forecasting across different spatial scales.
>
> **Q2**: explain in more depth the motivation for dataset
> > **I think it is necessary for the authors to explain in more depth the motivation for constructing such a long-term dataset, because as transportation infrastructure advances and human travel modes change, traffic data that is too ancient is not always helpful enough to understand future transportation patterns.**
>
> Our motivation for creating a extremely long-term dataset is to provide a robust basis for future traffic pattern analysis, which is essential for strategic planning in urban development and infrastructure investment. Our dataset, combined with our gap setting, allows for the assessment of traffic volumes over extended periods, enabling better planning for future road construction and traffic management. This extremely long-term perspective is crucial for aligning transportation infrastructure with future demands and ensuring the sustainability of urban environments.
>
> **Q3**: computational resources for the baselines experiments
> > **Due to the sheer size of the proposed dataset, it requires more computational resources. Therefore, whether this dataset can be popularized and whether peers can easily adopt it in their own research is also a concern. **
>
> Refer to your Question 1, our dataset is designed to filter out nodes with a time span exceeding 20 years, resulting in a relatively small spatial dimension that meets the criteria. Consequently, the dataset primarily has a longer temporal dimension, making the training cost acceptable. We have compiled the training time for one epoch of all the baselines on the largest dataset, PEMS04_gap, and the smallest dataset, tfNSW, as shown in the table below. (Additionally, we have included five traffic forecasting domain baselines: STGCN, ASTGCN, GWN, AGCRN, PDFormer according to Reviewer ytp5’s suggestion, as shown in Table 1 in the global comments.)
>
>
> **Table 2. Training time per epoch pf all the baselines in largest PEMS04_gap and smallest tfNSW**
>
> |baselines|Mamba|iTransformer|DLinear|Autoformer|Informer|FEDFormer|MICN|PatchTST|STGCN|ASTGCN|GWN|AGCRN|PDFormer|
> |-|-|-|-|-|-|--|-|-|-|-|-|-|-|
> |PEMS04_gap1|396.3|1310.2|237.9|1656.7|652.5|900.9|409.8|3649.4|2425.9|7921.9|4110.4|13369.1|8013.9|
> |tfNSW_gap1|13.7|44.1|7.4|148.2|73.3|380.1|101.2|55.3|37.3|228.8|50.4|658.6|47.9|

---

> > ### Author Response · Authors · 2024-11-28
> >
> > Dear reviewer 6qSU, we thank you again for your constructive review and comments. We hope we have addressed all your concerns. If you believe our comments and revisions have satisfied your concerns, would you please reconsider raising your score?

---

> > > ### Comment · Reviewer_6qSU · 2024-12-01
> > >
> > > Thanks for your detailed responses.  After reviewing the comments of other reviewers, I think my current score is appropriate.

---

> > > > ### Author Response · Authors · 2024-12-03
> > > >
> > > > Thanks for your response, we believe that we have addressed all your concerns, we have also extended some baselines as requested by reviewer ytp5. We’d greatly appreciate it if you could review our rebuttal and consider adjusting your scores based on our updates.

---

### Author Response · Authors · 2024-11-25
**Author Global Rebuttal**

We sincerely thank all the reviewers for the comments that have greatly improved the paper quality. We have taken all your comments seriously. We have added the new baselines, new analysis and clarify some confusing questions. The following global rebuttal provides 5 more baselines and computational statistics done during the rebuttal period.

- (1) For Reviewer ytp5’s suggestion, we have added five additional baselines shown in Table 1

**Table 1. Compasirons with different traffic forecasting baselines**
||||**STGCN**||**ASTGCN**||**GWN**||**AGCRN**||**PDFormer**||
|-|-|-|-|-|-|-|-|-|-|-|-|-|
|**Methods**|**Metrics**||**MSE**|**MAE**|**MSE**|**MAE**|**MSE**|**MAE**|**MSE**|**MAE**|**MSE**|**MAE**|
|||96|0.556|0.536|0.765|0.618|0.676|0.574|0.596|0.562|0.621|0.576|
||**1-yeargap**|192|0.562|0.539|0.764|0.626|0.581|0.546|0.565|0.543|0.545|0.536|
|||336|0.561|0.538|0.717|0.598|0.582|0.543|0.580|0.552|0.574|0.548|
|||96|1.256|0.884|1.441|0.959|1.250|0.875|1.168|0.866|1.256|0.899|
|**PEMS03\_gap**|**1.5-yeargap**|192|1.187|0.872|1.395|0.947|1.194|0.856|1.122|0.843|1.117|0.842|
|||336|1.167|0.853|1.273|0.877|1.1015|0.832|1.176|0.862|1.182|0.871|
|||96|2.059|1.176|2.236|1.234|1.987|1.148|1.950|1.150|2.068|1.189|
||**2-yeargap**|192|2.055|1.184|2.180|1.213|1.877|1.117|1.746|1.070|1.897|1.131|
|||336|2.022|1.169|1.889|1.117|1.847|1.107|1.930|1.140|1.982|1.161|
|||96|1.233|0.904|1.311|0.929|1.310|0.87|1.290|0.865|1.101|0.851|
||**1-yeargap**|192|1.398|1.025|1.443|1.023|1.255|0.879|1.318|0.907|1.078|0.816|
|||336|1.505|1.104|1.325|0.938|1.292|0.865|1.302|0.899|1.085|0.821|
|||96|1.369|1.082|1.350|0.937|1.684|1.063|1.222|0.961|1.097|0.849|
|**PEMS04\_gap**|**1.5-yeargap**|192|1.642|1.035|1.549|1.075|1.501|0.988|1.346|0.989|1.214|0.903|
|||336|1.368|0.862|1.199|0.832|1.584|0.997|0.231|0.963|1.130|0.854|
|||96|1.653|1.074|1.247|0.894|1.669|1.057|1.236|0.901|1.099|0.861
||**2-yeargap**|192|1.545|1.023|1.356|0.948|1.554|1.041|1.269|0.915|1.159|0.891|
|||336|1.556|1.011|1.174|0.842|1.128|0.869|1.314|0.947|1.032|0.816|
|||96|2.531|1.325|3.119|1.404|2.185|1.280|2.158|1.220|1.921|1.148|
||**1-yeargap**|192|2.560|1.337|2.405|1.276|2.223|1.311|2.144|1.213|2.248|1.260|
|||336|2.482|1.317|2.086|1.181|2.228|1.312|2.036|1.185|1.994|1.179|
|||96|1.772|1.084|2.370|1.192|1.309|0.896|1.166|0.808|1.664|1.055|
|**PEMS08\_gap**|**1.5-yeargap**|192|1.522|0.979|1.730|1.043|1.074|0.805|1.144|0.810|1.518|1.012|
|||336|1.495|0.966|1.444|0.948|1.533|0.999|1.078|0.784|1.049|0.801|
|||96|1.336|0.868|1.962|1.069|1.789|1.017|1.393|0.856|1.071|0.792|
||**2-yeargap**|192|1.218|0.819|1.292|0.795|1.007|0.761|1.229|0.813|1.171|0.826|
|||336|1.181|0.806|1.227|0.842|1.181|0.813|1.246|0.814|1.003|0.768|
|||96|1.176|0.776|1.404|0.874|1.451|0.871|1.283|0.824|1.070|0.756|
||**1-yeargap**|192|1.495|0.880|1.547|0.889|1.580|0.904|1.267|0.828|1.019|0.753|
|||336|1.517|0.901|1.617|0.910|1.534|0.892|1.048|0.786|0.822|0.710|
|||96|1.220|0.796|1.286|0.813|1.598|0.940|1.249|0.809|1.153|0.790|
|**TfNSW\_gap**|**1.5-yeargap**|192|1.624|0.934|1.615|0.902|1.620|0.950|1.228|0.808|1.095|0.792|
|||336|1.621|0.945|1.621|0.900|1.220|0.814|1.163|0.811|1.049|0.775|
|||96|1.201|0.785|1.331|0.823|1.414|0.862|1.124|0.749|0.969|0.702|
||**2-yeargap**|192|1.287|0.831|1.061|0.726|1.109|0.755|1.058|0.722|0.941|0.737|
|||336|1.260|0.810|0.996|0.693|1.011|0.695|0.960|0.749|0.917|0.715|

- (1) For Reviewer 6qSU’s suggestion, we have provided statistics on training times for the largest and smallest datasets shown in Table 2, demonstrating that our dataset and the settings we propose can be easily trained on one V100 GPU.

**Table 2. Training time for every epoch in the largest PEMS04_gap and smallest tfNSW**
||||**Mamba**||**iTransformer**||**DLinear**||**Autoformer**||**Informer**||**FEDFormer**||**MICN**||**PatchTST**||
|-|-|-|-|-|-|-|-|-|-|-|-|-|-|-|-|-|-|-|
|**Methods**|**Metrics**||**MSE**|**MAE**|**MSE**|**MAE**|**MSE**|**MAE**|**MSE**|**MAE**|**MSE**|**MAE**|**MSE**|**MAE**|**MSE**|**MAE**|**MSE**|**MAE**|
|||96|1.480|0.845|1.236|0.822|1.166|0.748|1.320|0.879|1.354|0.777|1.377|0.895|1.219|0.778|0.946|0.778|
||**1-yeargap**|192|1.543|0.867|1.088|0.733|1.184|0.758|1.599|0.961|1.229|0.757|1.312|0.890|1.262|0.799|1.026|0.821|
|||336|1.459|0.847|1.099|0.739|1.196|0.763|1.477|0.934|1.242|0.776|1.219|0.838|1.366|0.833|0.949|0.774|
|||96|1.745|0.944|1.355|0.877|1.372|0.846|1.658|1.016|1.615|0.879|1.440|0.919|1.204|0.778|1.025|0.807|
|**tfNSW\_gap**|**1.5-yeargap**|192|1.784|0.968|1.260|0.818|1.367|0.846|1.666|0.997|1.484|0.854|1.452|0.932|1.251|0.789|1.058|0.823|
|||336|1.628|0.939|1.284|0.835|1.385|0.854|1.515|0.945|1.500|0.883|1.346|0.871|1.203|0.770|0.995|0.788|
|||96|1.364|0.857|1.019|0.782|1.008|0.742|1.319|0.907|1.101|0.713|1.318|0.879|1.033|0.745|0.848|0.727|
||**2-yeargap**|192|1.195|0.783|0.884|0.701|1.024|0.749|1.319|0.891|1.054|0.721|1.222|0.855|1.094|0.737|0.909|0.757|
|||336|1.588|0.903|0.907|0.730|1.166|0.748|1.233|0.849|1.787|0.639|1.056|0.766|0.959|0.691|0.865|0.714|

---

### Comment · Area_Chair_FU7J · 2024-11-25
**Acknowledge the author responses**

Dear Reviewers,

Thank you very much for your effort. As the discussion period is coming to an end, please acknowledge the author responses and adjust the rating if necessary.

Sincerely,
AC

---

### Comment · Area_Chair_FU7J · 2024-11-28
**Discussion needed**

Dear Reviewers,

As you are aware, the discussion period has been extended until December 2. Therefore, I strongly urge you to participate in the discussion as soon as possible if you have not yet had the opportunity to read the authors' response and engage in a discussion with them. Thank you very much.

Sincerely,
Area Chair

---

### Note · Authors · 2025-02-24

**Comment:**

withdraw

**Withdrawal Confirmation:**

I have read and agree with the venue's withdrawal policy on behalf of myself and my co-authors.

---

### Meta-Review · Area_Chair_FU7J · 2024-12-19

**Metareview:**

This paper presents XXLTraffic with the longest timespan collected from Los Angeles, USA, and New South Wales, Australia, curated to support research in extremely long forecasting beyond test adaptation. The reviewers and I believe that providing this benchmark is a very meaningful contribution to the research community.  However, as some reviewers pointed out, the benchmark itself should have enough novelty to open or support a new research problem.   The reviewers feel that increasing the temporal range of the benchmark dataset may not be sufficient for that perspective.  I also agree with the reviewers' opinions.  When I submitted a benchmark paper to NeurIPS Datasets & Benchmarks track, I have often received similar comments.  Overall, considering the reviewers' opinions, I am sorry to recommend a reject.

**Additional Comments On Reviewer Discussion:**

Reviewers ytp5 and YRBw pointed out a lack of contribution, but the authors disagreed with their points.  I understand the perspectives both from the reviewers and authors.  I also believe that a benchmark paper should have novelty in the perspective of the problem that the dataset can be utilized.

---

### Decision · Program_Chairs · 2025-01-22

Reject